# HF-NeuS: Improved Surface Reconstruction Using High-Frequency Details

**Yiqun Wang**
KAUST

**Ivan Skorokhodov**
KAUST

**Peter Wonka**
KAUST

## Abstract

Neural rendering can be used to reconstruct implicit representations of shapes without 3D supervision. However, current neural surface reconstruction methods have difficulty learning high-frequency geometry details, so the reconstructed shapes are often over-smoothed. We develop HF-NeuS, a novel method to improve the quality of surface reconstruction in neural rendering. We follow recent work to model surfaces as signed distance functions (SDFs). First, we offer a derivation to analyze the relationship between the SDF, the volume density, the transparency function, and the weighting function used in the volume rendering equation and propose to model transparency as a transformed SDF. Second, we observe that attempting to jointly encode high-frequency and low-frequency components in a single SDF leads to unstable optimization. We propose to decompose the SDF into base and displacement functions with a coarse-to-fine strategy to increase the high-frequency details gradually. Finally, we design an adaptive optimization strategy that makes the training process focus on improving those regions near the surface where the SDFs have artifacts. Our qualitative and quantitative results show that our method can reconstruct fine-grained surface details and obtain better surface reconstruction quality than the current state of the art. Code available at `https://github.com/yiqun-wang/HFS`.

## 1 Introduction

3D reconstruction from a set of images is a fundamental challenge in computer vision [9]. In the recent past, the seminal framework NeRF [19] inspired a lot of follow up work by modeling 3D objects as a density function $\sigma(x)$ and view-dependent color $c(x, v)$ for each point $x \in R^3$ in the volume. The density function and view-dependent color function are implicit functions modeled by a neural network. The results of this approach are very strong and therefore NeRF inspired a large amount of follup up work, e.g. [18, 17, 24, 35, 20, 2].

In particular, one direction of work tries to constrain the density field to make it more consistent with a density field stemming from a surface. In the original formulation, almost arbitrary densities can be modeled by the neural network and there is no guarantee that a meaningful surface can be extracted from the density. Two noteworthy recent approaches, Neus [30] and VolSDF [32], proposed to embed a signed distance field in the volume rendering equation. Therefore, instead of modeling the density $\sigma$ with a neural network, these approaches model a signed distance function $f$ with a neural network. This leads to greatly improved surface reconstruction.

We build on this exciting recent work and seek further improvement in the quality of surfaces that are being reconstructed. To this end, we propose our method HF-NeuS consisting of three new building blocks. First, we analyze the relationship between the signed distance function on the one hand and the volume density, the transparency, and the weighting function on the other hand. We conclude from our derivation that it would be best to model a function that maps signed distances to the transparency and propose a class of functions that fulfill the theoretical requirements. Second, we observe that

36th Conference on Neural Information Processing Systems (NeurIPS 2022).

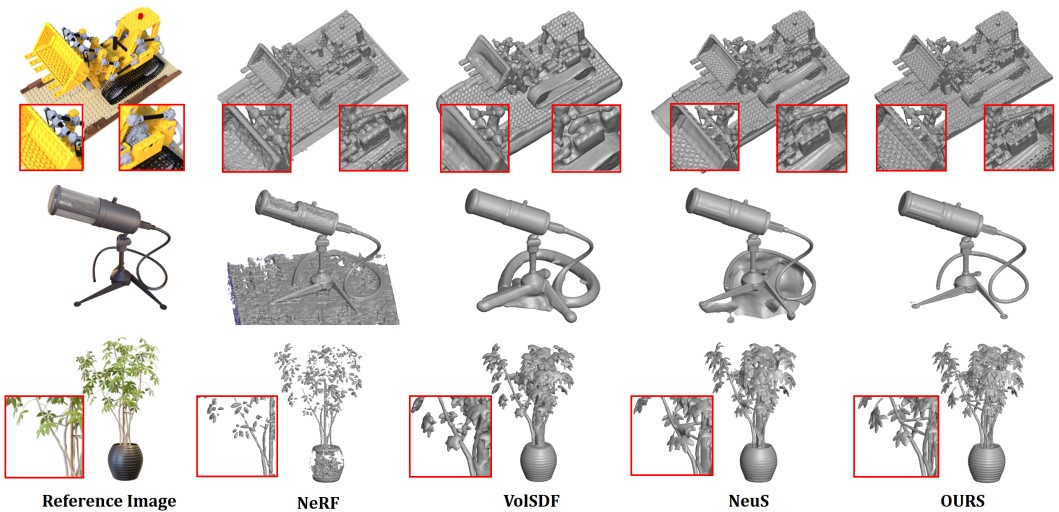

Figure 1: Qualitative evaluation on the Lego, Robot, and Ficus models. First column: reference images. Second to the fifth column: NeRF, VolSDF, NeuS, and OURS.

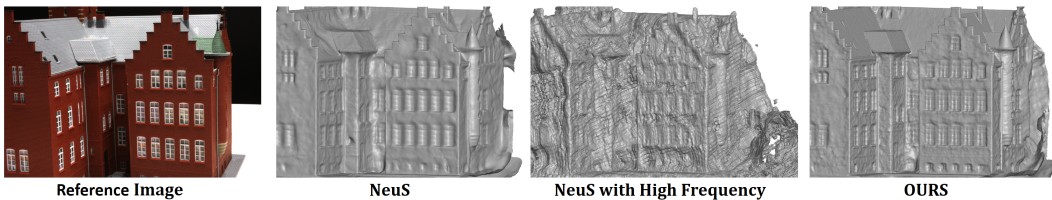

Figure 2: The challenge of using high-frequencies directly in the NeuS framework. First column: reference image. Second to the fourth column: NeuS, NeuS with high-frequency details, and OURS.

it is challenging to learn high-frequency details directly with a single signed distance function as shown in Fig. 2. We therefore propose to decompose the signed distance function into a base function and a displacement function following related work. We adapt this idea to the differentiable NeRF rendering framework and the NeRF training scheme. Third, the functions that translate distance to transparency can be chosen to have a parameter, which we call scale $s$. It controls the slope of the function (or the deviation of the derivative), which further controls the localization precision of the surface and how much out-of-surface colors influence the result. In previous work, this parameter $s$ is set globally but is trainable, so it can change from iteration to iteration. We propose a novel spatially adaptive weighting scheme to influence this parameter, so that the optimization focuses more on problematic regions in the distance field. The three building blocks are the three main contributions of the paper. In the results, we can see that HF-NeuS has a clear improvement in surface reconstruction. On the 15 scene DTU benchmark we can improve from the current best values of 0.87 (NeuS) and 0.86 (VolSDF) to 0.77 the Chamfer distance (See Figs. 1 and 4 for a visual comparison). The benchmark as well as the metric were proposed by previous work.

## 2   Related Work

**Multi-view 3D reconstruction.**  3D reconstruction based on multiple views is a fundamental challenge in the field of 3D vision. Classical 3D reconstruction algorithms usually reconstruct discrete 3D representations. The methods can be roughly categorized into voxel-based methods and point-based methods. Voxel-based methods [6, 27, 14, 4, 11, 22] first discretize the three-dimensional space uniformly into voxels, and then decide whether the surface occupies a particular voxel. Point-based methods [1, 7, 26, 25, 8] usually use the correlation between multiple views to reconstruct depth maps and fuse multiple depth maps into a point cloud. The point cloud needs to be

subsequently reconstructed into a mesh model using explicit algorithms like ball-pivoting [3] and Delaunay trianglulation [15] or implicit algorithms like Poisson surface reconstruction [13].

**Neural implicit surfaces.** Recently, neural implicit representations have received a lot of attention. The corresponding methods aim to reconstruct continuous implicit function representations of shapes directly from 2D images. A required building block is differentiable rendering, which maps the 3D scene representation to a 2D image for a given camera pose. DVR [21] utilizes surface rendering to model the occupancy function of a 3D shape, which uses a root search approach to obtain the location of the surface and predicts a 2D image. IDR [33] models the signed distance function of the shape and uses a sphere tracking algorithm to render 2D images. A significant milestone in 3D reconstruction was the development of NeRF [19]. It uses volume rendering to map a 3D density field and a 3D directional color field to a 2D image. The proposed representation is flexible enough so that realistic images can be synthesized. To model more complex scenes, NeRF++ [35] proposes to model the background scene with an additional neural radiance field, which handles the foreground and background separately, and achieves better results for large scenes. However, the density function is not as easy to control as the occupancy function or the signed distance function, and it is difficult to guarantee the smoothness of the generated 3D shape. Subsequently, UNISURF [23] embeds the occupancy function into the volume rendering equation of NeRF. They use a decay strategy to control which region to sample around the surface during training without explicitly modeling volumetric density. Using signed distance functions, VolSDF [32] embeds a signed distance function into the density formulation and proposes a sampling strategy that satisfies a derived error bound on the transparency function. NeuS [30] derive an unbiased density function equation using logistic sigmoid functions and introduce a learnable parameter to control the function's slope during rendering and sampling. Concurrent to our work, NeuralPatch [5] uses the homography matrix to warp the source patches adjacent to the reference image to constrain colors in the volume to come from closeby patches. However, the calculation of patch warping relies on the accurate surface normal, so it cannot be trained from scratch. Therefore, it is only used as a fine-tuning or post-processing method for other algorithms to optimize the surface. We consider VolSDF and NeuS as the current state of the art and we will compare to these two methods.

**High-frequency detail reconstruction.** It is generally difficult for neural networks to learn high-frequency information from raw signals. Inspired by the field of natural language processing, positional encoding [19, 29] is used to guide the network to reconstruct high-frequency details. Positional encoding spreads the original signal into different frequency bands using sine and cosine functions of different frequency. Subsequently, SIREN [28] proposes to use the sin function as activation function in the network. MipNeRF [2] presents an integrated positional encoding to control frequency in different scales. Park *et al.* [24] proposed to use a coarse-to-fine learning strategy to gradually increase high-frequency information, which was subsequently used for pose estimation [17]. Hertz *et al.* [10] further propose a spatially adaptive progressive coding strategy. For surface reconstruction, implicit displacement fields were proposed for single-view 3D reconstruction [16]. Based on the supervision of ground truth SDF values of sampled points, the method utilizes separate networks to model the base SDF and implicit displacement fields. Subsequently, Wang *et al.* [34] utilize the SIREN network to learn the base implicit function and implicit displacement function, respectively, for point cloud reconstruction tasks. In contrast to our proposed algorithm, these methods require 3D supervision. Further, they do not involve the NeRF formulation or volume rendering. In our work, we build on these ideas to develop a new state-of-the-art algorithm for multi-view reconstruction.

## 3 Method

As input we consider a set of $N$ images $I = \{I_1, I_2...I_N\}$, and their corresponding intrinsic and extrinsic camera parameters $\Pi = \{\pi_1, \pi_2...\pi_N\}$. HF-NeuS aims to reconstruct the representation of 3D surface $S$ as implicit functions. Specifically, we encode surfaces as signed distance fields. We will explain our method in three parts: 1) First, we show how to embed the signed distance function into the formulation of volume rendering and discuss how to model the relationship between distance and transparency. 2) Then, we propose to utilize an additional displacement signed distance function to add high-frequency details to the base signed distance function. 3) Finally, we observe that the function that maps signed distances to transparency is controlled by a parameter $s$ that determines the slope of the function. We propose a scheme to set this parameter $s$ in a spatially varying manner

depending on the gradient norm of the distance field, rather than keeping it constant for the complete volume within a single training iteration.

## 3.1 Modeling transparency as transformed SDF

We first review the integral formula for volume rendering and derive a relationship between transparency and the weighting function (the product between density and transparency). Based on this analysis, we discuss the criteria for functions that are suitable to map signed distances to transparency and propose a class of functions that fulfill the theoretical requirements.

Given a ray $\mathbf{r}(t) = \mathbf{o} + \mathbf{td}$, the volume rendering equation is used to calculate the radiance $C$ of the pixel corresponding to the ray $\mathbf{r}$. The volume rendering equation is an integral along the ray and involves the following quantities defined for each point in the volume: the volume density $\sigma$ and the (directional) color $\mathbf{c}$. In addition, the volume has compact support and the boundaries of the volume are encoded by $t_n$ and $t_f$.

$$C(\mathbf{r}) = \int_{t_n}^{t_f} T(t)\sigma(\mathbf{r}(t))\mathbf{c}(\mathbf{r}(t), \mathbf{d})dt \tag{1}$$

The transparency $T(t)$ is derived from the volume density as explained below. The function $T(t)$ denotes the accumulated transmittance along the ray from $t_n$ to $t$

$$T(t) = \exp\left(-\int_{t_n}^{t} \sigma(\mathbf{r}(s))ds\right), \tag{2}$$

and $T(t)$ is a monotonic decreasing function with a starting value of $T(t_n) = 1$. The product $T(t)\sigma(\mathbf{r}(t))$ can be regarded as a weighting function $w(t)$ in the volume rendering equation as in Eq. (1).

In order to involve a signed distance function $f$, we have to define a function $\Psi$ to transform a signed distance function so that it can be used to compute the density related terms in the rendering equation. One way is to directly model a density function $\sigma(\mathbf{r}(t)) = \Psi(f(\mathbf{r}(t)))$ as proposed by VOLSDF [32]. Taking this approach, a sampling method is required to satisfy an error bound of the sampling to make it less than an error threshold by gradually reducing the scale parameter. Another way is to model the weighting function $w((t)) = \Psi(f(\mathbf{r}(t)))$ as proposed by NeuS. The NeuS paper showcases a complex derivation to get the expression for the density function $\sigma$.

We rethink this problem to obtain a simplified derivation by focusing on transparency instead of the weighting function and also a better understanding of the problem, as follows:

$$\frac{d(T(t))}{dt} = -T(t)\sigma(\mathbf{r}(t)) \tag{3}$$

An interesting observation is that the derivative of the transparency function $T'(t)$ is the negative weighting function. The weighting function has the property of having a maximum on the surface. We take the derivative of the weighting function and set it to 0 to find the extrema (maxima), as follows.

$$\frac{d(T(t)\sigma(\mathbf{r}(t)))}{dt} = -\frac{d^2(T(t))}{dt^2} = -\frac{d(T'(t))}{dt} = 0 \tag{4}$$

Assuming a planar surface and a single ray-plane intersection, we can see that the extremum point, denoted as $t_s$, of the weighting function $w(t)$ will also be the extremum point of the derivative of the transparency function $T'(t)$. The point $t_s$ is expected to be the intersection of the ray and the surface. Therefore, we consider defining the transparency function directly as $T(t) = \Psi(f(\mathbf{r}(t)))$. If the transparency function is designed in such a way that its derivative $T'(t)$ reaches a minimum on the surface, it follows that the weighting function has a maximum on the surface. Therefore, one can directly model a transparency function under the condition that its derivative has a minimum on the surface. This is conceptually simpler than modeling the weighting function $w(t)$ as proposed by NeuS. We compute the derivative of $\Psi(f(\mathbf{r}(t)))$ as follows.

$$\frac{d(\Psi(f(\mathbf{r}(t))))}{dt} = \Psi'(f(\mathbf{r}(t)))\frac{df}{d\mathbf{r}}\frac{d\mathbf{r}}{dt} = \Psi'(f(\mathbf{r}(t)))\nabla f(\mathbf{r}(t)) \cdot \mathbf{d} \tag{5}$$

where $\nabla f(\mathbf{r}(t)) \cdot \mathbf{d}$ is the product of the surface normal and the ray direction, which is a constant in case of a planar surface and a single ray-plane intersection. The signed distance function is zero

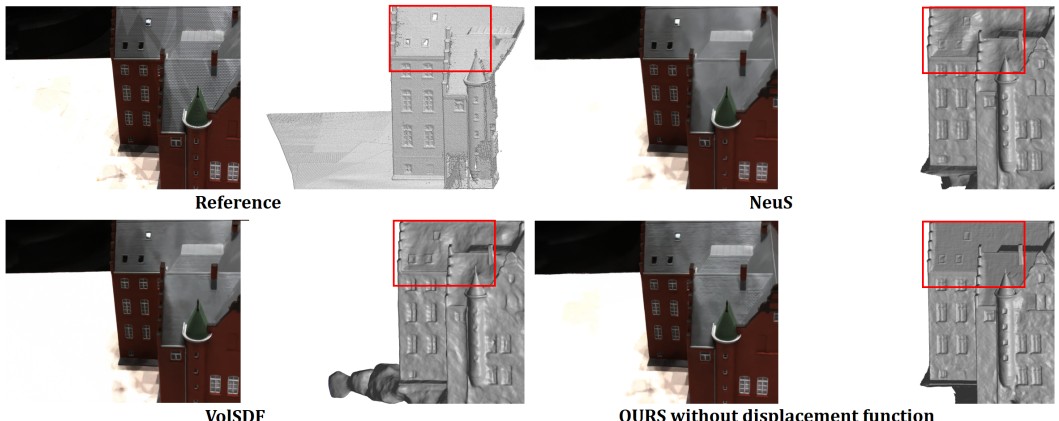

Reference

NeuS

VolSDF

OURS without displacement function

Figure 3: Comparing NeuS and VolSDF with our transparency model. Ground truth is on the top left. For each method, the left shows the reconstructed image and the right the reconstructed surface.

on the surface. Hence $\Psi'$ has an extremum at $f = 0$. This also means $\Psi$ has the steepest slope at the surface of the shape. On the other hand, the signed distance function is positive outside of the object, and negative when entering the interior of the object. We generally assume that $t = t_n$ is outside so that the signed distance starts positive and decays to a negative value along a ray, which is a monotonic decreasing function. According to the characteristics of transparency $T(t) = \Psi(f(\mathbf{r}(t)))$, the transparency starts at 1 at $t = t_n$ and is a monotonic decreasing function to 0 inside the object. This inverse property results in the $\Psi$ function being a monotonic increasing function from 0 to 1. Therefore, we have our design criteria for $\Psi$: $\Psi$ should be a monotonic increasing function from 0 to 1, with the steepest slope at 0.

A very intuitive idea to satisfy this criteria is to use a sigmoid function and normalize the function to have an output in the interval $[0, 1]$. We simply use the logistic sigmoid function proposed by NeuS [30] for a fair comparison. However, our idea is more general and other sigmoid functions could be used. Our designed transparency function is as follows,

$$T(t) = \Psi_s(f(\mathbf{r}(t))) = \frac{1}{1 + e^{-sf(\mathbf{r}(t))}}, \tag{6}$$

where $\Psi_s$ is the logistic sigmoid function with parameter $s$ controlling the slope of the function. Note that the parameter $s$ is also the standard deviation of the function $\Psi'_s$. We will use this fact later when discussing the adaptive version of the framework.

Given the differentiable transparency function $T(t)$, the volume density $\sigma$ can be easily calculated following Eq. 3.

$$\sigma(\mathbf{r}(t)) = -\frac{T'(t)}{T(t)} \tag{7}$$

For discretization, we bring Eq. 5 and Eq. 6 into Eq.7, and take advantage of the properties of the derivative of the logistic sigmoid function $\Psi'_s = s\Psi_s(1 - \Psi_s)$. We can get the $\sigma$ formula for the discretization computation:

$$\sigma(\mathbf{r}(t_i)) = s\left(\Psi_s(f(\mathbf{r}(t_i))) - 1\right)\nabla f(\mathbf{r}(t_i)) \cdot \mathbf{d} \tag{8}$$

Then the volume rendering integral can be approximated using $\alpha$-composition, where $\alpha_i = 1 - exp(-\sigma_i(t_{i+1} - t_i))$. For multiple surface intersections, we follow the same strategy as NeuS [30], where $\alpha_i = clamp(\alpha_i, 0, 1)$. Compared with NeuS, we obtain a simpler formula for the density $\sigma$ for the discretization computation, reducing the numerical problems caused by division in NeuS. Furthermore, our approach does not need to involve two different sampling points, namely section points and mid-points, which makes it easier to satisfy the unbiased weighting function. Since there is no need to calculate the SDF and the color separately for the two different point sets, the color and the geometry are more consistent compared to NeuS. Compared to VolSDF [32], since the transparency function is explicit, our method can use an inverse distribution sampling computed with the inverse CDF to satisfy the approximation quality. Thus no complex sampling scheme as in VolSDF is required. A visual comparison is shown in Fig. 3.

## 3.2 Implicit displacement field without 3D supervision

In order to enable a multi-scale fitting framework, we propose to model the signed distance function as a combination of a base distance function and a displacement function [34, 16] along the normal of the base distance function. The implicit displacement function is an additional implicit function. The reason for this design is that it is difficult for a single implicit function to learn low-frequency and high-frequency information at the same time. The implicit displacement function can complement the base implicit function, so that it is easier to learn high-frequency information.

Compared with the task of learning implicit functions from point clouds, reconstructing 3D shapes from multiple images makes it more difficult to learn high-frequency content. We propose to use neural networks to learn frequencies at multiple scales, and to gradually increase the frequency content in a coarse-to-fine manner.

Suppose $f$ is the combined implicit function that represents the surface we want to obtain. The function $f_b$ is the base implicit function that represents the base surface. Following [34], the displacement implicit function $f_{d'}$ is used to map the point $x_b$ on the base surface to the surface point $x$ along the normal $n_b$ and vice versa $f_d$ is used to map the point $x$ on the base surface to the surface point $x_b$ along the normal $n_b$, thus $f_{d'}(\mathbf{x}_b) = f_d(\mathbf{x})$. Because of the nature of implicit functions, the relationship between the two functions can be expressed as follows,

$$f_b(\mathbf{x}_b) = f(\mathbf{x}_b + f_{d'}(\mathbf{x}_b)\,\mathbf{n}_b) = 0 \tag{9}$$

where $\mathbf{x}_b = \frac{\nabla f_b(\mathbf{x}_b)}{\|\nabla f_b(\mathbf{x}_b)\|}$, is the normal of $\mathbf{x}_b$ on the base surface. To compute the expression for the implicit function $f$, we bring the formula $\mathbf{x}_b = \mathbf{x} - f_{d'}(\mathbf{x}_b)\,\mathbf{n}_b$ into the Eq. (9) and obtain the expression for the combined implicit function:

$$f(\mathbf{x}) = f_b(\mathbf{x} - f_d(\mathbf{x})\,\mathbf{n}_b) \tag{10}$$

Therefore, we can use the base implicit function and the displacement implicit function to represent the combined implicit function. However, two challenges arise. First, the Eq. 10 is only satisfied if the point $x$ is on the surface. Second, the normal at the point $\mathbf{x}_b$ is difficult to estimate when only knowing the position $\mathbf{x}$. We rely on two assumptions to solve the problem. One assumption is that this deformation can be applied to all iso-surfaces, i.e. $f_b(\mathbf{x}_b) = f(\mathbf{x}_b + f_{d'}(\mathbf{x}_b)\,\mathbf{n}_b) = c$. In this way the equation is assumed to be valid for all points in the volume and not only on the surface. Another assumption is that $\mathbf{x}_b$ and $\mathbf{x}$ are not too far away, thus $\mathbf{n}_b$ can be replaced with normal $\mathbf{n}$ on the point $\mathbf{x}$ in the Eq. (10). We control the magnitude of the implicit displacement function using a displacement constraint $4\Psi_s'(f_b)$.

To precisely control the frequency, we use positional encoding to encode the base implicit function and the displacement implicit function separately. We would like to note some differences to [34]. We use positional encoding instead of Siren [28], so that the frequency can be explicitly controlled by a coarse-to-fine strategy rather than simply using two Siren networks with two different frequency levels. This is useful when 3D supervision is not given. More details are shown in the supplementary. Positional encoding decomposes the input position $\mathbf{x}$ into multiple selected frequency bands.

$$\gamma(\mathbf{x}) = [\gamma_0(\mathbf{x}), \gamma_1(\mathbf{x}), ..., \gamma_{L-1}(\mathbf{x})] \tag{11}$$

where each component consists of a $\sin$ and a $\cos$ function with different frequency.

$$\gamma_j(\mathbf{x}) = \left[\sin\left(2^j \pi \mathbf{x}\right), \cos\left(2^j \pi \mathbf{x}\right)\right] \tag{12}$$

Directly learning high-frequency positional encoding makes the network susceptible to noise, because wrongly learned high-frequencies hinder the learning of low frequencies. This problem is less pronounced if 3D supervision is available, however high-frequency information of images is easily introduced into the surface generation as noise. We use the coarse-to-fine strategy proposed by Park *et al.* [24] to gradually increase the frequency of the positional encoding.

$$\gamma_j(\mathbf{x}, \alpha) = \omega_j(\alpha)\,\gamma_j(\mathbf{x}) = \frac{(1 - \cos\left(clamp\left(\alpha L - j, 0, 1\right)\pi\right))}{2}\gamma_j(\mathbf{x}) \tag{13}$$

where $\alpha \in [0, 1]$ is the parameter to control the frequency information involved. In each iteration, $\alpha$ is increased by $1/n_{\max}$ until it touches 1, where $n_{\max}$ is the maximum number of iterations.

We utilize two kinds of positional encoding $\gamma(\mathbf{x}, \alpha_b), \gamma(\mathbf{x}, \alpha_d)$ with different parameter $\alpha_b$ and $\alpha_d$. We set $\alpha_b = 0.5\alpha_d$ and only control $\alpha_d$ for simplicity. We also use two MLP functions $MLP_b, MLP_d$ for fitting the base and displacement functions.

$$f(\mathbf{x}) = MLP_b(\gamma(\mathbf{x}, \alpha_b) - 4\Psi_s'(f_b)MLP_d(\gamma(\mathbf{x}, \alpha_d))\,\mathbf{n}), \tag{14}$$

where $\mathbf{n} = \frac{\nabla f_b(\mathbf{x})}{\|\nabla f_b(\mathbf{x})\|}$ that can be computed by the gradient of $MLP_b$ and $\Psi_s'(f_b) = \Psi_s'(MLP_b(\gamma(\mathbf{x}, \alpha_b)))$. The $s$ of the displacement constraint should be clamped during training. We show how to control the adaptive $s$ in the supplemental materials.

We bring this implicit function into Eq. (6) for calculating the transparency so that the radiance (color) $\hat{C}_s$ of images can be computed by the volume rendering equation.

To train the network, we employ the loss function $\mathcal{L} = \mathcal{L}_{rad} + \mathcal{L}_{reg}$, which includes the radiance loss and the Eikonal regularization loss of the signed distance functions. For the regularization loss, we constrain both the base implicit function and the detailed implicit function.

$$\mathcal{L} = \frac{1}{M} \sum_s \left\| \hat{C}_s - C_s \right\|_1 + \frac{1}{N} \sum_k \left[ (\|\nabla f_b(\mathbf{x}_k)\|_2 - 1)^2 + (\|\nabla f(\mathbf{x}_k)\|_2 - 1)^2 \right] \tag{15}$$

### 3.3   Modeling an adaptivate transparency function

In previous subsections, the transparency function is parametrized as a sigmoid function controlled by the scale $s$. This parameter controls the slope of the sigmoid function and it is also the standard deviation of the derivative. We can also say that it controls the smoothness of the function. When $s$ is large, the value of the sigmoid function drops sharply as the position moves away from the surface. On the contrary, the value decreases smoothly when $s$ is small. However, choosing a single parameter $s$ per iteration gives the same behavior at all spatial locations in the volume.

Since two signed distance functions need to be reconstructed, especially after the high frequency is superimposed, it is easy to break the Eikonal constraint, i.e., make the SDF's gradient norm deviate from 1 in some positions. Even with the regularization loss, it is impossible to avoid this problem.

We propose to use the gradient norm of the signed distance field to weight the parameter $s$ in a spatially varying manner, increasing $s$ when the gradient norm along the ray direction is larger than 1. The intuition is that the implicit function with the larger gradient norm undergoes more abrupt changes, which indicates a region that should be improved. Making $s$ larger in such regions makes the distance function more precise by magnifying its errors, especially near the surface. In order to adaptively modify the scale $s$, we propose the following equation:

$$T(t) = \left( 1 + e^{-s\exp\left(\sum\limits_{i=1}^{K} \omega_i \|\nabla f_i\| - 1\right) f(\mathbf{r}(t))} \right)^{-1}, \tag{16}$$

where $\nabla f$ is the gradient of the signed distance function, and $K$ is the number of sampling points, $\omega_i$ is the normalized $\Psi_s'(f_i)$ as the weight and $\sum\limits_{i=1}^{K} \omega_i = 1$.

While this method can be used to control the transparency function, it can also be used for the hierarchical sampling stage proposed by standard NeRF [19]. By locally increasing $s$, more samples will be generated near the surface where the signed distance values change more rapidly. This mechanism also helps to optimization to focus on these regions in the volume.

## 4   Experiments

**Baselines.** We compare HF-NeuS to the following three state-of-the-art baselines: (1)NeuS [30] is the most relevant baseline for our work. We consider it to be the best published method. (2)VolSDF [32] is concurrent work to NeuS. We consider it to be the second best published method. Overall it also performs very well. (3)NeRF focuses on image synthesis and is included for completeness. NeRF

Table 1: Quantitative results on the DTU dataset.

| Metric | Method | 24 | 37 | 40 | 55 | 63 | 65 | 69 | 83 | 97 | 105 | 106 | 110 | 114 | 118 | 122 | **Mean** |
|---|---|---|---|---|---|---|---|---|---|---|---|---|---|---|---|---|---|
| Fidelity | NeRF | 1.90 | 1.60 | 1.85 | 0.58 | 2.28 | 1.27 | 1.47 | 1.67 | 2.05 | 1.07 | 0.88 | 2.53 | 1.06 | 1.15 | 0.96 | 1.49 |
| | VOLSDF | 1.14 | 1.26 | 0.81 | 0.49 | 1.25 | 0.70 | 0.72 | 1.29 | 1.18 | **0.70** | 0.66 | **1.08** | 0.42 | 0.61 | 0.55 | 0.86 |
| | NeuS | 1.37 | **1.21** | 0.73 | 0.40 | 1.20 | 0.70 | 0.72 | **1.01** | 1.16 | 0.82 | 0.66 | 1.69 | 0.39 | **0.49** | 0.51 | 0.87 |
| | OURS | **0.76** | 1.32 | **0.70** | **0.39** | **1.06** | **0.63** | **0.63** | 1.15 | **1.12** | 0.80 | **0.52** | 1.22 | **0.33** | **0.49** | **0.50** | **0.77** |
| PSNR | NeRF | 26.24 | 25.74 | 26.79 | 27.57 | 31.96 | 31.50 | 29.58 | 32.78 | 28.35 | 32.08 | 33.49 | 31.54 | 31.0 | 35.59 | 35.51 | 30.65 |
| | VOLSDF | 26.28 | 25.61 | 26.55 | 26.76 | 31.57 | 31.50 | 29.38 | 33.23 | 28.03 | 32.13 | 33.16 | 31.49 | 30.33 | 34.90 | 34.75 | 30.38 |
| | NeuS | 28.20 | 27.10 | 28.13 | 28.80 | 32.05 | 33.75 | 30.96 | 34.47 | 29.57 | 32.98 | 35.07 | 32.74 | 31.69 | 36.97 | 37.07 | 31.97 |
| | OURS | **29.15** | **27.33** | **28.37** | **28.88** | **32.89** | **33.84** | **31.17** | **34.83** | **30.06** | **33.37** | **35.44** | **33.09** | **32.12** | **37.13** | **37.32** | **32.33** |

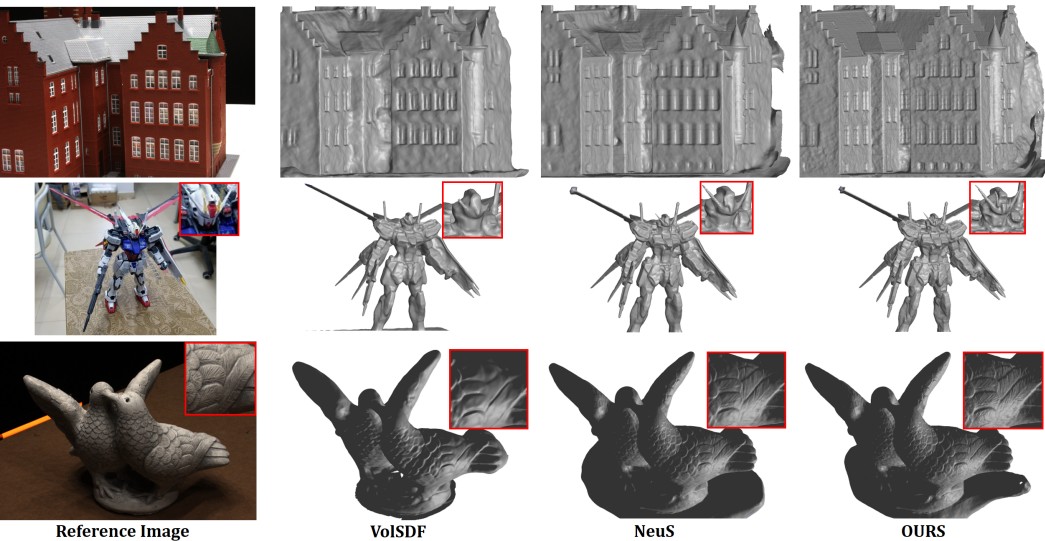

Figure 4: Qualitative evaluation on DTU (first and third rows) and BlendedMVS (second row).

is not really a surface reconstruction method and does not reconstruct high-quality surfaces, but it is very good in image-based metrics. We use a threshold of 25 (as proposed by NeuS [30]) to extract surfaces for the comparisons. For all three methods, we use the default parameters and the number of iterations recommended in their respective papers. We do not include older methods in the comparison, such as UNISURF [23] or IDR [33], because NeuS and VolSDF have better results.

**Datasets.** We conduct experiments on the DTU dataset [12]. We follow previous work and choose the same 15 models for comparison. DTU is a multi-view stereo dataset. Each scene consists of 49 or 64 views with $1600 \times 1200$ resolution. We further choose 9 challenging scenes from other datasets: 6 scenes from the NeRF-synthetic dataset [19] and 3 scenes from BlendedMVS [31](CC-4 License). The image resolution of NeRF synthetic dataset [19] is $800 \times 800$ and 100 views are provided for each scene. The dataset contains objects with very obvious detailed and sharp features, such as the Lego and Microphone scenes. We chose this dataset for the analysis of reconstructions of high-frequency details. The BlendedMVS dataset is similar to the DTU dataset, but with a richer background. This dataset provides image resolution of $768 \times 576$. We also select models with high-frequency details or sharp features which are difficult to reconstruct. In all three datasets, ground truth surfaces and camera poses are provided.

**Evaluation metrics.** To evaluate the quality of the reconstruction, we follow previous work and used Chamfer distance (lower values are better) and PSNR (higher values are better). For the DTU dataset, we use the official evaluation protocol, which means computing the mean of accuracy (distance from the reconstructed surface to the ground truth surface) and completeness (distance from the ground truth surface to the reconstructed surface). For DTU and BlendedMVS, the background is not part of the ground truth surface. Therefore, we remove the background for computing the Chamfer distance, following previous work. The NeRF-synthetic dataset [19] has no background, so we only remove disconnected parts for all competing methods.

Table 2: Quantitative results on NeRF-synthetic and BlendedMVS datasets.

| Metric($10^{-2}$) | Method | Chair | Ficus | Lego | Materials | Mic | Ship | **Mean** | Bread | Dog | Robot | **Mean** |
|---|---|---|---|---|---|---|---|---|---|---|---|---|
| Fidelity | NeRF | 2.12 | 5.17 | 3.05 | 1.51 | 4.77 | 3.54 | 3.36 | 0.102 | 0.693 | 2.325 | 1.07 |
| | VOLSDF | 1.26 | 1.54 | 2.83 | 1.35 | 3.62 | 2.92 | 2.37 | 0.074 | 0.354 | 1.453 | 0.63 |
| | NeuS | 0.74 | 1.21 | 2.35 | 1.30 | 3.89 | 2.33 | 1.97 | 0.068 | 0.173 | 1.036 | 0.43 |
| | OURS | **0.69** | **1.12** | **0.94** | **1.08** | **0.72** | **2.18** | **1.12** | **0.065** | **0.155** | **0.922** | **0.38** |
| PSNR | NeRF | **33.00** | **30.15** | **32.54** | 29.62 | **32.91** | **28.34** | **31.09** | 31.27 | 27.46 | 25.33 | 28.02 |
| | VOLSDF | 25.91 | 24.41 | 26.99 | 28.83 | 29.46 | 25.65 | 26.86 | 31.05 | 28.24 | 25.46 | 28.25 |
| | NeuS | 27.95 | 25.79 | 29.85 | 29.36 | 29.89 | 25.46 | 28.05 | 31.32 | 28.71 | 25.87 | 28.63 |
| | OURS | 28.69 | 26.46 | 30.72 | **29.87** | 30.35 | 25.87 | 28.66 | **31.89** | **29.42** | **26.15** | **29.15** |

Table 3: Ablation study results.

| Datasets | Chamfer Distance | | | | | | PSNR | | | | | |
|---|---|---|---|---|---|---|---|---|---|---|---|---|
| | Base | Base+H | Base+C2F | IDF+H | IDF+C2F | FULL | Base | Base+H | Base+C2F | IDF+H | IDF+C2F | FULL |
| DTU | 1.08 | 1.20 | 1.07 | 1.25 | 0.89 | 0.78 | 31.77 | 32.73 | 32.56 | 32.69 | 32.13 | 32.49 |
| NeRF-Synthetic | 2.51 | 3.61 | 2.95 | 2.83 | 1.35 | 0.91 | 28.39 | 30.52 | 30.63 | 30.12 | 29.88 | 30.31 |
| BlendedMVS | 0.43 | fail | 0.63 | 0.47 | 0.41 | 0.38 | 28.63 | fail | 27.35 | 28.20 | 28.95 | 29.15 |

**Implementation details.** We use MLPs to model two signed distance functions $f_b$ and $f_d$. Each MLP consists of 8 layers. Related work like NeuS [30] and IDR [33] also use MLPs with 8 layers. We use Adam with learning rate $5e^{-4}$ for the network training using NVIDIA TITAN A100 40GB graphics cards. For adaptive sampling, we first uniformly sample 64 points on the ray, then calculate the SDF and its gradient at these points. We utilize the Eq. 16 to calculate the gain of the $s$ parameter, and then adaptively update the weight according to the gain and sample an additional 64 points. For the coarse-to-fine strategy, we observe that using $\alpha_d^0 = 0$ at the beginning for surface reconstruction produces smoothed results. We utilize $\alpha_d^0 = 0.5$ and $\alpha_b^0 = 0.5\alpha_d^0 = 0.25$ for both signed distance functions. We set $L = 16$ for the parameter of the frequency band of positional encoding. For other parameter settings, please see the supplemental materials.

**Comparison.** In table 1, we show quantitative results with other competitors on 15 scenes of the DTU dataset [12]. The values shown in the upper part of the table measure the fidelity of the surface reconstruction, the Chamfer distance. The numbers indicate that HF-NeuS significantly outperforms NeRF. In most scenes, HF-NeuS is better than VolSDF and NeuS so that the overall average distance is also improved. In the lower part of the table we show the PSNR values. It can be seen that our PSNR surpasses all other methods. We further compare the visual quality achieved by different methods. As shown in Fig. 4, HF-NeuS can reconstruct high-frequency details. For example, the windows have better geometric details, and the feathers of the bird are more distinct.

Most of the scenes in the DTU dataset have smooth surfaces, and high-frequency details are not obvious. We selected 9 challenging models from the NeRF-synthetic dataset [19] and BlendedMVS dataset [31], which have more high-frequency details. For example, the Lego model has uneven repeating bumps, and the power cord of the Mic model has a very thin structure (Fig. 1). The robot model has richer edge and corner features (Fig. 4 second row). As shown in Table 2, the gap between our surface reconstruction quality and that of all other methods widens. This shows that HF-NeuS is especially advantageous for surface reconstructions with high-frequency information. We can also observe that NeRF is very good in the image-based metric (PSNR) while performing poorly in the surface reconstruction metric (Chamfer distance). This observation is consistent with previous work. Compared with the NeRF-Synthetic dataset, the BlendedBMS dataset has a more complex background, this also restricts the performance of NeRF to a certain extent. Besides outperforming other baselines in terms of quantitative error, we also achieve better results in terms of qualitative visual effects. As shown in Fig. 1, HF-NeuS can more accurately reconstruct the details of each Lego block and even some of the tiny holes that are not reconstructed by any other method. For the Robot scene, HF-NeuS can reconstruct more accurate facial contours and sharper horns. Finally, for the Mic model, HF-NeuS can clearly reconstruct the power cord, while other methods will mess up this structure.

**Ablation study.** We verify the influence of different modules on the reconstruction results, including the coarse-to-fine module, the implicit displacement function module, and the position-adaptive $s$

control module. In Table 3, "Base "refers to the baseline method, which is NeuS. "H" means we use high-frequency positional encoding. Here we set L=16 to represent high frequencies. "C2F" refers to the coarse-to-fine optimization strategy with high-frequency positional encoding. We set the initial $\alpha$ to 0.5. "IDF" represents using the implicit displacement function in reconstruction. For each dataset, we chose the mean of the three scenes as the quantitative metric. From the results of the BlendedMVS dataset, we can observe that the divergence of network training can be prevented based on the coarse-to-fine strategy. From the DTU and NeRF-synthetic datasets, introducing high-frequency directly can easily lead to overfitting on these datasets. This means that an increase in PSNR cannot guarantee the improvement of the fidelity of surface reconstruction. Although the coarse-to-fine module can alleviate this mismatch to some degree, it is difficult to further improve the performance. However, adding the implicit displacement function component improves the fidelity of the surface reconstruction and PSNR at the same time. During reconstruction, the network with adaptive $s$ can help to improve the reconstruction quality upon more complex scenes.

**Limitation.** As shown in Fig. 5, our method still has challenges. We show a reference ground truth image, our corresponding reconstructed image, and our reconstructed surface. For the grid of ropes of the ship, some overfitting to ground-truth radiance is still observed. Specifically, the grid of ropes is visible in the image, but the surface is not reconstructed accurately. Another limitation is that the individual thin ropes are missing. We also visualize a bad case of Table 1 where the error is larger than that of the other methods as shown in Fig. 14 DTU Bunny in the supplementary material. In this case, the lighting of this model varies and the texture is not as pronounced, thus it is difficult to reconstruct the details of the belly. Further, integrating our proposed IDF increases computation time.

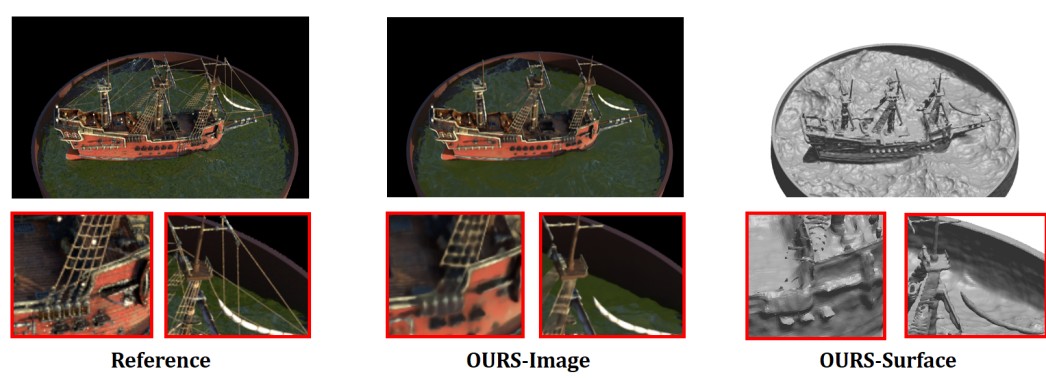

Figure 5: Limitation. First column: the reference ground truth images. Second column: our synthetic images. Last column: our reconstructed surface.

## 5   Conclusion

We introduce HF-NeuS, a new method for multi-view surface reconstruction with high-frequency details. We propose a new derivation to explain the relationship between signed distance and transparency and propose a class of functions that can be used. By decomposing the signed distance field into a combination of two independent implicit functions, and using adaptive scale constraints to focus on optimizing the regions where the implicit function distribution is not ideal, a more refined surface can be reconstructed compared to previous work. The experimental results show that the method outperforms the current state of the art in terms of quantitative metrics and visual inspection. An interesting direction for future work is to explore the reconstruction of scenes under different lighting modalities. Finally, we do not expect negative social impacts that will be directly linked to our research. Negative social impacts of surface reconstruction in general are possible though.

## Acknowledgements

We would like to acknowledge support from the SDAIA-KAUST Center of Excellence in Data Science and Artificial Intelligence and the NSFC No.62202076.

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
