# A  Additional results

## A.1  Frequency ablation study

We perform an ablation study on the coarse-to-fine parameter $\alpha_d$ and the number of frequency bands $L$. In Fig. 6, we show the surface reconstruction results of the DTU Buddha model under different frequency parameters. Each model is trained for 300K iterations. In the first row we show the results of surface reconstruction quality under different coarse-to-fine parameters $\alpha_d$. It can be seen that when the parameter is too small, the surface reconstruction tends to be oversmoothed. When the parameter is too large, many artifacts will appear in the reconstruction results. We adopt $\alpha_d = 0.5$ for our model. In the second row we show the effect of different numbers of frequency bands on the reconstruction results when the coarse-to-fine parameter $\alpha_d$ is fixed. We can observe that many details such as the cracks of the Buddha are not included in the reconstruction results when the number of frequency bands is too small. When the number of frequency bands is too large, although the details increase, high-frequency noise is also introduced, which will cause the reconstructed model to be unnatural. We adopt $L = 16$ as the number of frequency bands of our model.

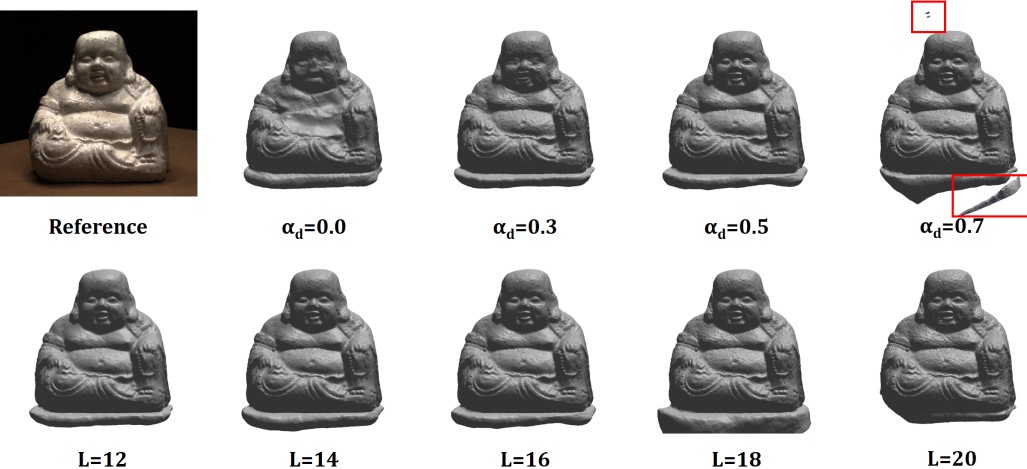

Figure 6: Frequency ablation study. First row: the reference image and the reconstructed surface using different coarse-to-fine parameters $\alpha_d$ with $L = 16$. Second row: the reconstructed surface using different numbers of frequency bands $L$ with $\alpha_d = 0.5$.

## A.2  Adaptive transparency ablation study

In Fig. 7, an ablation experiment for adaptive transparency is shown. We compare the reconstruction results without using the adaptive transparency strategy. We selected the Lego model from the NeRF-synthetic dataset, the skull model (scan65) from the DTU dataset and the dog model from the BlendedMVS for validation. For the Lego model in the first column, we find that the adaptive transparency strategy can better deal with regions with holes. Holes can be better unclogged during reconstruction. For the skull model in the second column, since there are only a few training images on the right side of the skull, the cheekbones are difficult to reconstruct accurately, but our method can better reconstruct these regions. For the dog model in the third column, because the belt is a fine part, it is not easy to be reconstructed well by the network. The strategy using adaptive transparency can better focus on these regions and reconstruct surface details better.

## A.3  Additional reconstruction results

In this section, more qualitative results are provided. Fig. 8 and Fig. 9 show additional comparisons with NeuS [30] and Volsdf [32] on the DTU dataset, the NeRF-Synthetic dataset, and the BlendedMVS dataset.

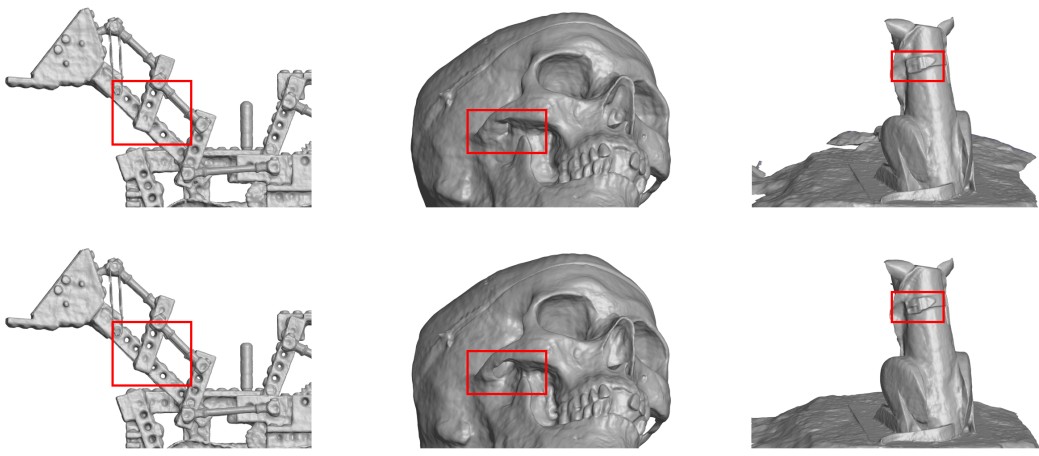

Figure 7: Adaptive transparency ablation study. First row: the reconstructed surface without the adaptive transparency strategy. Second row: the reconstructed surface using adaptive transparency.

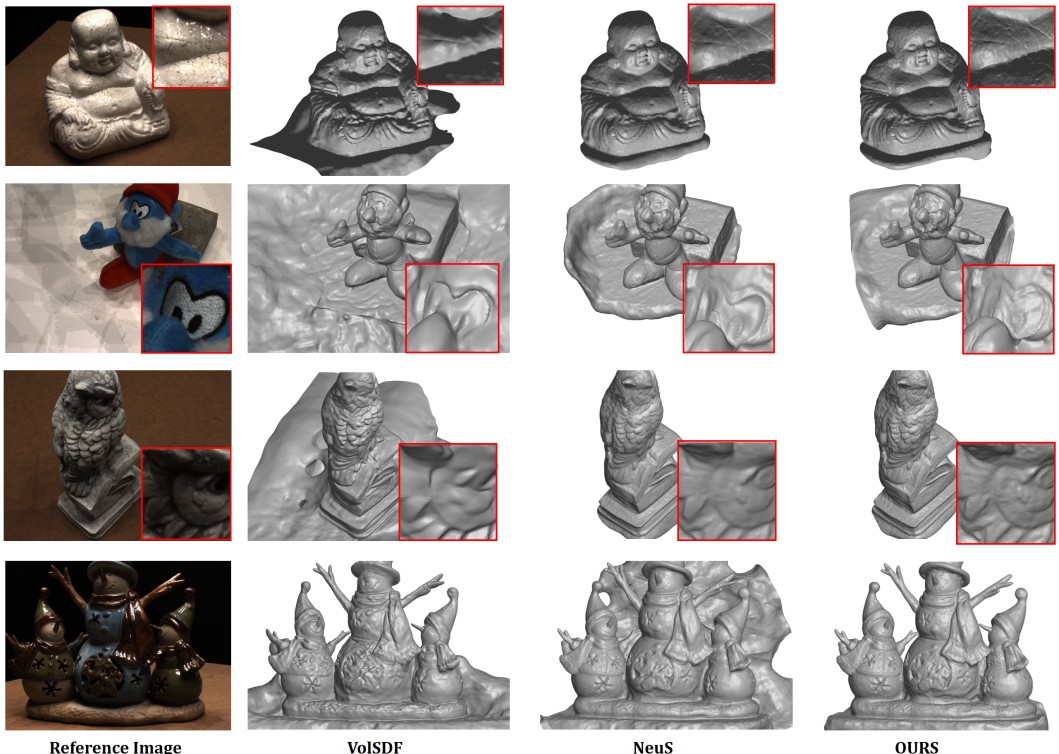

|  Reference Image | VolSDF | NeuS | OURS |

Figure 8: Additional reconstruction results on the DTU dataset. First column: reference images. Second to the fourth column: VolSDF, NeuS, and OURS.

## A.4 Qualitative comparison of different design choices

**Benefits of modeling transparency.** We provide a qualitative comparison to Volsdf and NeuS in Fig. 10. Our transparency model is easy to evaluate, avoids numerical problems due to division, and does not need to sample section points and mid-points separately. "OUR Base-Sigmoid" shows better geometry consistency on the roof of the house compared with VolSDF and NeuS. We also conduct an experiment to show the results with different choices of transparency $\Psi_s$ in Fig. 10. "OUR Base-Laplace" means using the CDF of the Laplace distribution for $\Psi_s$. In the negative semi-axis,

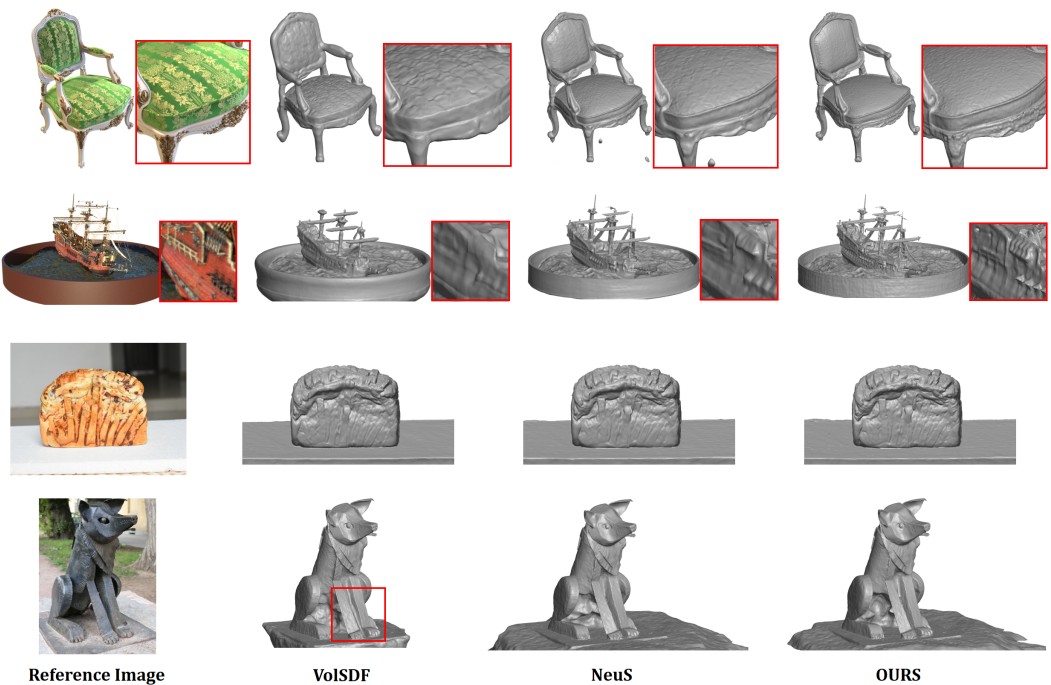

| Reference Image | VolSDF | NeuS | OURS |

Figure 9: Additional reconstruction results on the NeRF-Synthetic dataset and the BlendedMVS dataset. First column: reference images. Second to the fourth column: VolSDF, NeuS, and OURS.

the derivative of the Laplace distribution CDF divided by the distribution is a constant function, the scale parameter $s$, and this will affect the quality of surface reconstruction.

**Benefits of using positional encoding.** We provide a qualitative comparison to the result of using Siren [28] network in Fig. 10. We observed that the IDF using Sirens ("OURS-Siren" in Fig. 10) used in [34] can obtain a high PSNR result but low geometry fidelity. Although [34] also uses a coarse-to-fine strategy between two frequency levels, we found that the method still has problems when learning high-frequency details because of the high-frequency noise involved at the beginning. Our IDF using positional encoding does not use high-frequency information at the beginning of training, which makes the training more stable. In general, We provide a solution that allows more fine-grained control over frequency. This approach is more stable for the case without 3D supervision.

**Ablations of using different Eikonal regularization.** We provide an ablation study of Eikonal regularization in Fig. 10. We observe that training without the regularization of the base SDF ("OUR-w/o Base Reg") results in slightly worse reconstruction quality. Thus constraining the base SDF can help improve the quality of the reconstruction.

**Training with few images.** We conduct an experiment for surface reconstruction with 10% of the training images in Fig. 11. We find that our method can keep the structure of reconstructed objects complete compared to NeuS, and can better reconstruct parts such as thin stripes with fewer training images. PSNR of NeuS and OURS is 28.31 and 31.77 respectively for the shown test image, which is also improved.

## A.5 Evaluating the use of the displacement function

We provide a visualization of an example result in Fig. 12. As can be seen from the figure, the base SDF can reconstruct a smooth model of the Buddha. The displacement function is used to add extra details like cracks in the Buddha model and some small holes in the forehead (Base + IDF).

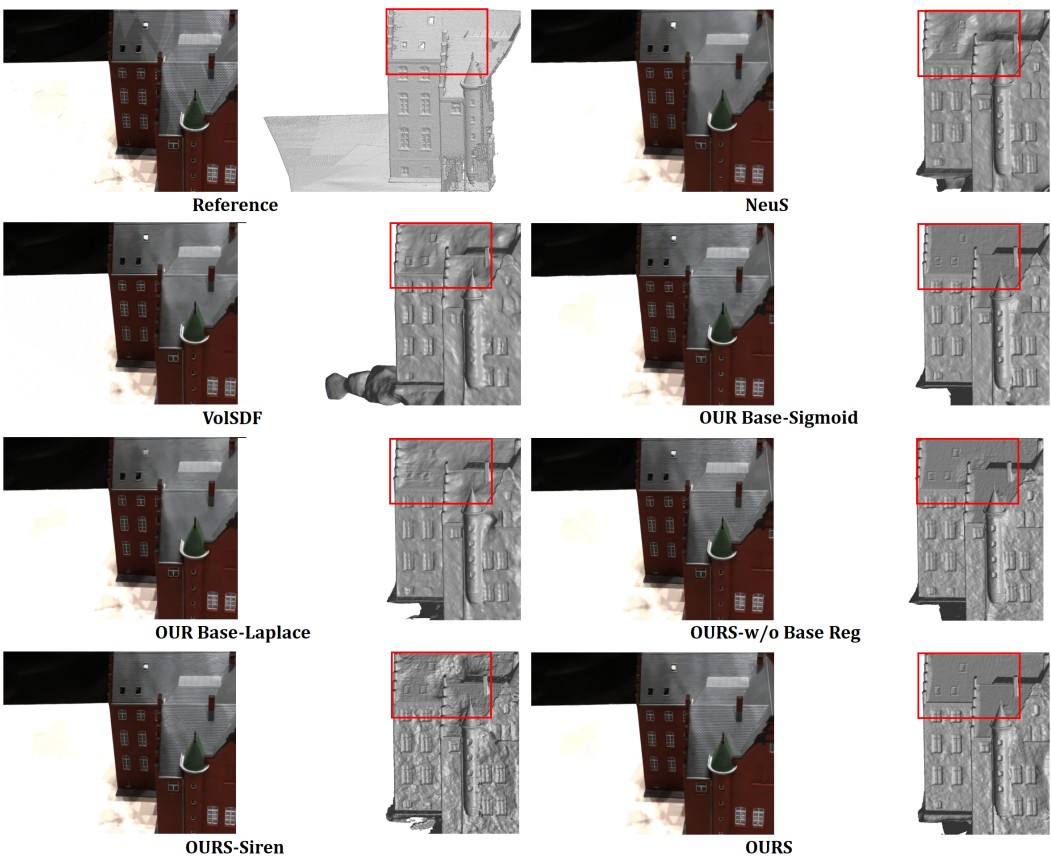

Figure 10: Qualitative comparison of different algorithm design choices. On the top left we show a ground truth image and the corresponding geometry. For each method in the comparison, we show the reconstructed image on the left and the reconstructed surface on the right.

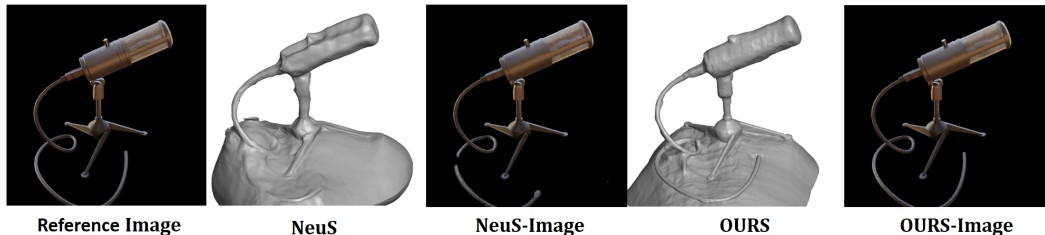

Figure 11: Qualitative evaluation of training with few images. The first is reference image. Second to the fifth: NeuS, the synthetic image of NeuS and OURS, the synthetic image of OURS.

## A.6 Qualitative ablation study

We show qualitative results for different ablations in Fig. 13. We observed that learning high-frequency details (Base+H) using only image information is difficult, and the network may overfit the image without reconstructing the correct geometry. Learning high-frequency details using IDF (IDF+H) will alleviate the noise but still produce large geometric errors. We provide a solution that allows more fine-grained control over frequency. By using a coarse-to-fine positional encoding, the frequency can be explicitly controlled by a coarse-to-fine strategy. This approach is more stable for the case without 3D supervision. Compared with only using the coarse-to-fine strategy (Base+C2F), IDF using C2F with positional encoding (IDF+C2F) has the ability to further improve the geometric fidelity as well as the image reconstruction quality as measured by PSNR. Using an adaptive strategy(FULL) can improve geometric fidelity and PSNR even further.

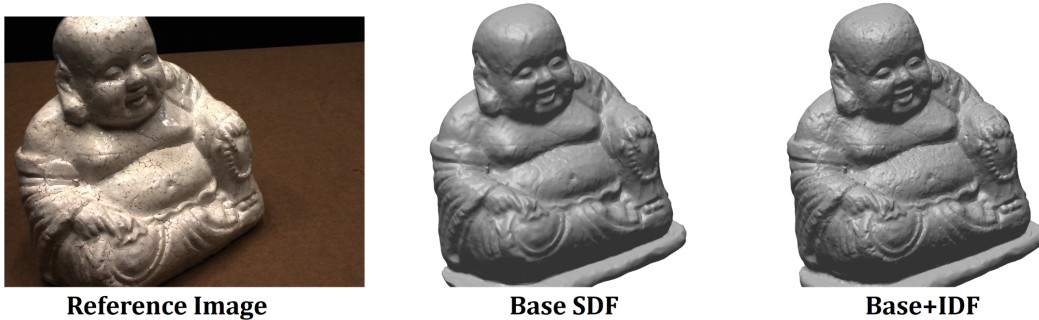

**Reference Image**  **Base SDF**  **Base+IDF**

Figure 12: The visualization of the SDF decomposition. The left is the referene image, the middle is base SDF, the right is the combined SDF.

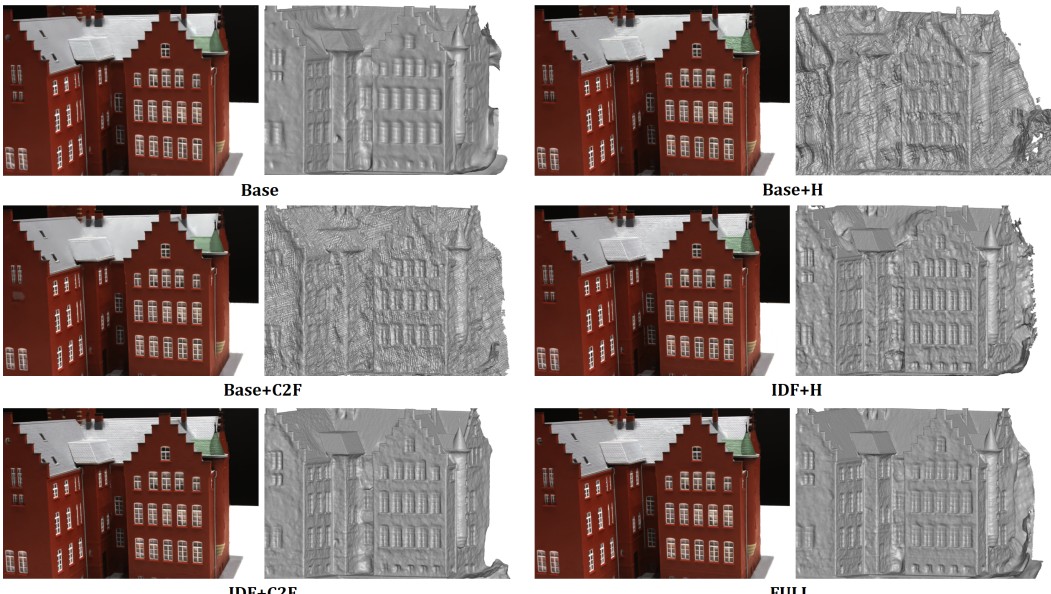

Base    Base+H

Base+C2F    IDF+H

IDF+C2F    FULL

Figure 13: Qualitative evaluation of the ablation study. For each setting, the left is the synthetic images of the different methods, and the right is the reconstructed surfaces

### A.7  The visualization of local errors

We provide a heatmap result for local errors in Fig. 14 to better highlight the local error. It can be seen that we have a higher improvement in the details, such as the roof and the details in the shovel of the excavator.

## B  Additional implementation details

### B.1  The computation of the Chamfer distance

Given are two point clouds $S_1$ and $S_2$ densely sampled from a surface. The Chamfer distance is defined as follows.

$$d_{Chamfer}(S_1, S_2) = \frac{1}{2}\left(\frac{1}{|S_1|}\sum_{p \in S_1}\min_{q \in S_2}\|p - q\|_2^2 + \frac{1}{|S_2|}\sum_{q \in S_2}\min_{p \in S_1}\|p - q\|_2^2\right) \quad (17)$$

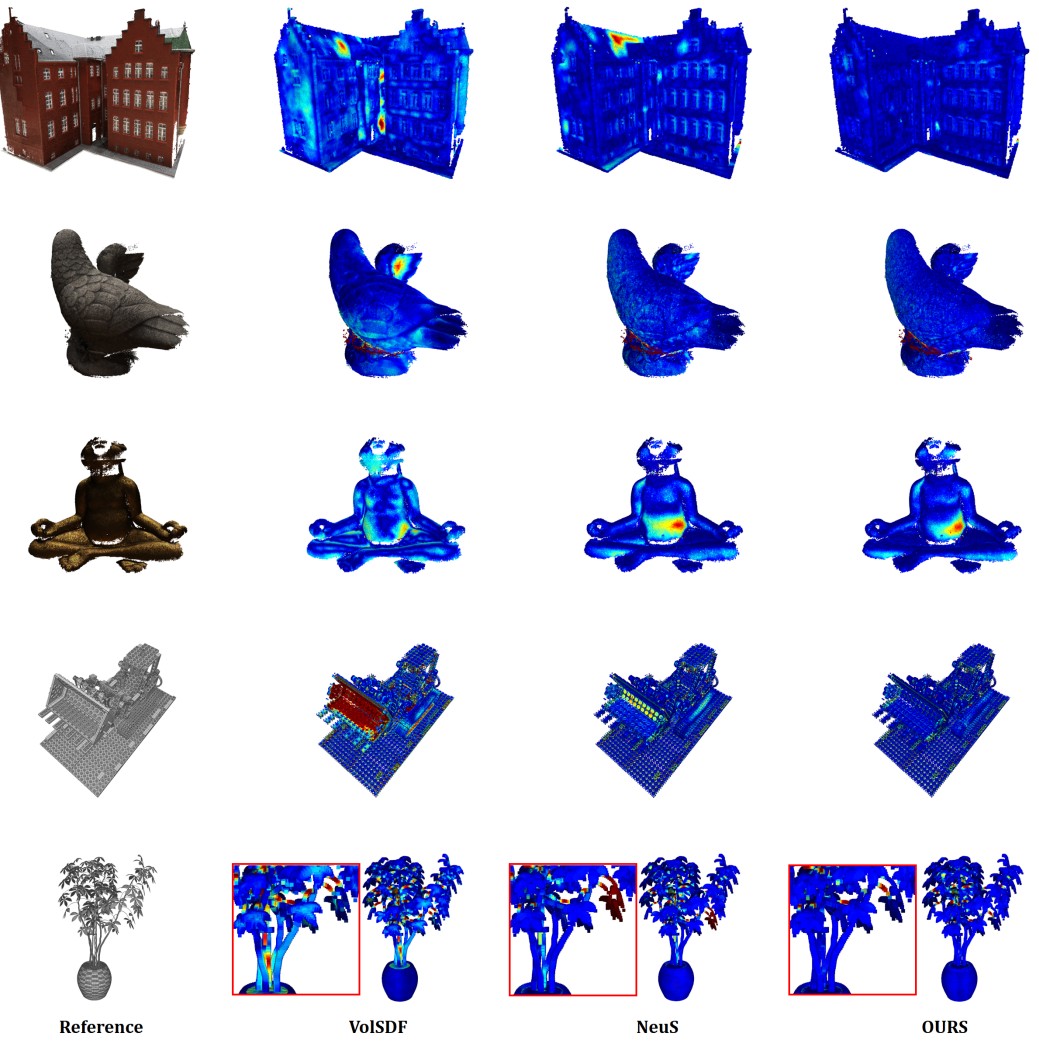

| Reference | VolSDF | NeuS | OURS |

Figure 14: The visualization of local errors. First column: reference images. Second to the fourth column: VolSDF, NeuS, and OURS.

## B.2 Training and inference time

The training time of each scene is around 20 hours for 300k iterations. The inference time for extracting a mesh surface with high resolution (512 grid resolution for marching cubes) is around 60 seconds and rendering an image at the resolution of 1600x1200 takes around 540 seconds.

## B.3 Displacement constraint

Without supervision in 3D space, it is difficult to obtain accurate base implicit functions and implicit displacement functions simultaneously for the complete volume. Specifically, the lack of supervision is more problematic far from the surface since the volume rendering is mainly influenced by the region in space close to the surface. We assume that the displacement function $f_d$ approaches 0 as $\mathbf{x}$ moves away from the surface. In regions far away from the surface, the details are not needed because we do not extract a surface in these regions. Therefore, when the point is very far away, we assume that the value of the base function and the combined function is the same. This assumption reduces the difficulty of network training and suppresses the effects of high-frequency noise when 3D supervision is not present.

The derivative of the sigmoid function $\Psi'_s$ has the property of converging to 0 away from the surface when applied to the signed distance function. It can therefore be used to constrain displacement distances. In practice, we use the function to constrain the displacement function $f_d$ as follows.

$$f(\mathbf{x}) = f_b(\mathbf{x} - 4\Psi'_{(0.01s)}(f_b)f_d(\mathbf{x})\,\mathbf{n}) \tag{18}$$

We relax the constraints near the surface with a factor of 0.01 and the $s$ of the displacement constraint is clamped to less than $10^3$. Because $\Psi'_s = s\Psi_s(1 - \Psi_s)$, the $\Psi'_s$ is small over all the space with small $s$ at the beginning, which can be interpreted as having a tight constraint at the beginning to speed up the convergence of base SDF. As the number of iterations increases, the constraint on the displacement function near the surface is gradually relaxed to fit high-frequency details.

## B.4  Hierarchical sampling

We use hierarchical sampling to determine the sampling points on the ray. We first uniformly sample 64 points, and then sample a new set of 64 points according to the weighting function. The weighting function can be seen as a probability density function, where the probability of sampling is high when the ray intersects the surface, and the probability is low elsewhere. The scale parameter $s$ controls if the sample points are mainly located very close to the surface, or if they spread out around the surface. We also weight the parameter $s$ of rays at different spatial locations according to the gradients. We first calculate the signed distance function $f_i$ and its gradient $\nabla f_i$ using 64 uniformly sampled points, and then the weighting coefficients $c$ for scale $s$ are calculated.

$$c = \exp\left(\sum_{i=1}^{K} \overline{\Psi'_s}(f_i)\|\nabla f_i\| - 1\right) \tag{19}$$

where $\overline{\Psi'_s}(f_i) = \Psi'_s(f_i)/\sum_{i=1}^{K} \Psi'_s(f_i)$ is the normalized weight for each point on the ray, and $K$ is the number of sampling points. We modulate the magnitude of the scale $s$ with the coefficient $c$, and use scale $s$ to control the probability density for the adaptive sampling. Here we do not clamp $s$. As $s$ increases, our coefficient $c$ depends more on the gradients close to the surface.