# OpenReview forum: "HF-NeuS: Improved Surface Reconstruction Using High-Frequency Details"
_NeurIPS.cc/2022/Conference — NeurIPS 2022 Accept_

### Official Review · Reviewer_YNsW · 2022-06-17

**Rating:** 7
**Confidence:** 5
**Soundness:** 3 good
**Presentation:** 3 good
**Contribution:** 3 good

**Summary:**

The paper improves the detail representation of geometric details for implicit surface reconstruction using multiview images.
It contains 3 contributions:

1. an analysis of the SDF and values in the volumetric rendering, which leads to formally deriving the suitable transformation function to use SDF for volumetric rendering.
2. an improved implicit displacement field for better detail reconstruction, which contains several changes to the original implicit displacement field to robustly work with 2D supervision
3. a strategy to improve the optimization by adapting the mapping function in (1) spatially.

**Questions:**

1. Regarding the displacement constrain $\Psi'_s$, in the supplemental, the authors mentioned using some progressive strategy to gradually increase the constraint. If I understand correctly, this means gradually requiring the displacement to be spatially sparse and more focused on the surface area. The motivation is not very well explained. Shouldn't I want the opposite - by requiring a tighter spatial bound around the base surface at the beginning of the training, the base SDF can quickly converge, so that when optimizing the displacement $f_d$, $f_b$ remains stable and hopefully the composed function $f$ converges faster. Why is this not the case?
2. Can I understand the adaptive transparency function described in (15) as a form of hard example mining? What do you think $\|\nabla f\|$ would be a good indicator? BTW, should it be $\\|\nabla f\\|$ instead of $\|\nabla f\|$?
3. Table 3: Is X+C2F adding C2F to X+H?  FULL is adding adaptive transparency to IDF+H+C2F? If this is not the case, then the IDF+adaptive s is missing


**Limitations:**

Yes.

**Strengths And Weaknesses:**

# Strengths:
This paper is very well written and shows impressive improvements compared to the state-of-the-art for reconstructing geometric details.
The three steps are well motivated and the ablation shows that they all contribute to the improvement. The entirety of the paper provides a good basis for NeRF-based multi-view 3D reconstruction.

# Weakness:
I think while this paper was very clear in stating the 3 contributions (section 3.1-section 3.3), it's not very clear what exactly is novel (proposed by the paper) and what is from prior work. This is particularly true in section 3.2. L175-201 seem to be purely prior work [32], which should be clarified and condensed. More effort should be put into explaining and motivating the differences.

There is a small section in the supplemental (B.2) explaining hierarchical sampling based on $cs$. But I don't see any result demonstrating the effectiveness of this sampling strategy.

---

> ### Author Response · Authors · 2022-08-02
> **Comments**
>
> We thank the reviewer for the insightful and detailed review. We respond to each question in the following.
>
> **R5-Q1. It's not very clear what exactly is novel (proposed by the paper) and what is from prior work. More effort should be put into explaining and motivating the differences.**
>
> Please refer to ALL-Q1 and ALL-Q2.
>
> **R5-Q2. I don't see any result demonstrating the effectiveness of this hierarchical sampling strategy. Table 3: Is X+C2F adding C2F to X+H? FULL is adding adaptive transparency to IDF+H+C2F? If this is not the case, then the IDF+adaptive s is missing.**
>
> As you discussed, X+C2F is adding C2F to X+H and FULL is adding adaptive transparency to IDF+H+C2F.  We model an adaptive transparency function which is implemented by modifying the scale $s$ adaptively. And the scale $s$ is used in the volume rendering integral and hierarchical sampling. We present the qualitative results in Table 4, and the visual comparison in Fig. 11 in the additional material as also requested by other reviewers. It can be seen that this strategy can improve surface reconstruction quality and repair undesired parts.
>
> **R5-Q3. Explanation of the displacement constraint.**
>
> Thanks for the explanation you presented. We found that the previous explanation did not take into account the change in $s$
> Because $\Psi'_s =s\Psi_s(1-\Psi_s)$, the $\Psi'_s$  is small over all the space with small $s$ at the beginning, which can be interpreted as having a tight constraint at the beginning and then releasing the constraint near the surface. We changed the explanation in the supplemental material in the revision.
>
> **R5-Q4. Can I understand the adaptive transparency function described in (15) as a form of hard example mining? What do you think $\left\| {\nabla {f}} \right\|$  would be a good indicator?**
>
> Yes, the adaptive transparency function is like a form of hard example mining, which focuses on low-quality regions of the SDF in the spatial domain.
> For the signed distance field, the norm of the gradient is expected to be 1. If the norm is greater than 1, it means that the signed distance at that location changes more drastically, which means that the change of the signed distance function is larger in a shorter distance.
> We increase the number of sampling points in these regions to better improve the surface quality.
>
> **R5-Q5. Should it be $\left|| {\nabla {f}} \right||$ instead of $\left| {\nabla {f}} \right|$?**
>
> Thank you for the suggestion, it is $\left|| {\nabla {f}} \right||$ instead of $\left| {\nabla {f}} \right|$ in the equation, we will fix the typo in the revision.

---

> ### Comment · Reviewer_YNsW · 2022-08-05
> **Thank you for the response**
>
> My questions have been sufficiently addressed in the author's response and some suggestions have been already integrated in the revision.
>
> I stand with original my rating and recommend accepting the paper.

---

### Official Review · Reviewer_iTh1 · 2022-06-24

**Rating:** 5
**Confidence:** 4
**Soundness:** 3 good
**Presentation:** 3 good
**Contribution:** 3 good

**Summary:**

The paper present s new method for surface reconstruction using the recently popular neural volume rendering approach. The authors suggest improving existing methods, which often lack high-frequency details. To do so, the authors present various changes. First, they propose transforming a learned sign distance function into the transparency function, rather than the physical density or the weighting function as done in relevant baselines. Additionally, following previous works, the authors decompose the sign distance function into base and displacement functions, accompanied by a gradually increasing frequency learning strategy. Lastly, they propose improving certain regions by weighing based on the properties of the sign distance function. The paper presents reconstruction results compared to existing baselines at three different datasets.

**Questions:**

Except for the mentioned concerns raised in the previous section, I would appreciate it if the authors would answer the following questions:

1. Could the authors explain their choice of the logistic sigmoid function for Psi?
2. (Relates to the above) Have the authors considered using the Gaussian or Laplace distributions CDF as the transformation function Psi? Why would one choice will be better than the other?
3. I suspect equation 9 has a typo, and it should be f_d(x_b)? If not, please explain the transition from equation 8.
4. The intention behind equation 15 is unclear. Could the authors please explain?

**Limitations:**

The authors addressed the method limitations. However, I encourage the authors to move the limitations section to the main paper in the revised version.

**Strengths And Weaknesses:**

Strengths:

1. The need for high-frequency details improvement is well motivated and understood.
2. I find some of the suggested improvements innovative and contributive to future works. More specifically, I like the idea of transforming the transparency and examining its derivate and relation to the weighting function, and the idea of the adaptive weighting strategy. I mostly appreciate the concept of weighting based on the eikonal equation satisfaction.
3. The related work section is concrete, well written, and covers the paper area for newcomers.

Weaknesses:

1. The contribution of each component is not straightforward. Although the authors conducted an ablation study, they have not presented qualitative results or discussed the potential implications of each component for the overall improvement.
2. Following but separately from the above, I’m concerned with the need for the base and displacement decomposition. Because this decomposition is not unique, and since high frequencies encoding adds noise, more possible wrongly reconstructions exist that overfit the rendering. I believe that a visualization of the decomposition would emphasize its benefits. Another alternative is to show an ablation using the decomposition (“IDF”) with frequencies as the baseline NeuS.
3. There is a clear difference between the presented VolSDF baseline results shown in this paper and other resources. I direct the authors to the original VolSDF paper for results of the DTU and BlendedMVS datasets (figures 5 and 10), and to the recent work RefNeRF (https://dorverbin.github.io/refnerf/) for the synthetic NeRF results (see video). I would appreciate it if the authors could explain this difference.
4. There are some missing methodology implementation details. For example, it is unclear what are the discretization formulas for the transparency and the volume rendering integral approximations, given a set of samples along the ray. Also, I believe the authors should state how multiple surface intersections are handled (line 173). Additionally, it is not certain how the displacement constraint is imposed, nor its relation to equation 13.
5. The superiority of the presented methods is mostly given quantitively but not qualitatively. It is also apparent in the PSNR results, surpassing all other methods, but the paper does not show rendering comparisons. Similar to the ablation study, I believe that additional qualitative results (even in the supplementary) would help the paper statements.

---

> ### Author Response · Authors · 2022-08-02
> **Comment**
>
> We thank the reviewer for the insightful and detailed review. We respond to each question in the following.
>
> **R4-Q1. No rendering results showing ablation study and comparisons and discussing each component.**
>
> As requested, we show the qualitative results of each component in Fig. 11 of the revised supplementary material. We observed that learning high-frequency details (Base+H) using only image information is difficult, and the network may overfit the image without reconstructing the correct geometry.
> Learning high-frequency details using IDF (IDF+H) will alleviate the noise but still produce large geometric errors. We provide a solution that allows more fine-grained control over frequency. By using a coarse-to-fine positional encoding, the frequency can be explicitly controlled by a coarse-to-fine strategy. This approach is more stable for the case without 3D supervision. Compared with only using the coarse-to-fine strategy (Base+C2F), IDF using C2F of positional encoding (IDF+C2F) has the ability to further improve the geometric fidelity with the PSNR increasing. Using an adaptive strategy(FULL) can further improve geometric fidelity while improving PSNR.
> We also provide the qualitative results (synthetic image and surface) in Fig. 8 of the supplementary material and show better surface reconstruction results compared with other methods.
>
> **R4-Q2. Because this decomposition is not unique, and since high frequencies encoding adds noise, more possible wrong reconstructions exist that overfit the rendering. I believe that a visualization of the decomposition would emphasize its benefits.**
>
> We provide a visualization in Fig. 10 in the revised supplementary material as requested. As can be seen from the figure, the base SDF can reconstruct a smooth model of the Buddha. The displacement function is used to add extra details like cracks in the Buddha model and some small holes in the forehead.
>
> **R4-Q3. The difference between the presented VolSDF baseline results and other resources like RefNeRF.**
>
> We actually adopt the official VolSDF code. Our quantitative results are consistent with previous work, e.g. the VolSDF paper. However, the method is heuristic and rerunning the method many times and manually selecting the best version will give better results. For example, we noticed that by running the VolSDF multiple times, one can achieve better results for some of the models, specifically for the DTU house and ficus model. However, this gives an advantage to VolSDF as this process cannot be done automatically. We nevertheless updated the paper with the improved versions of these two models.
> For the RefNeRF, they do not provide results of reconstructed surfaces but normal maps. The normal map is also weighted by volume rendering and does not fully represent the level set. In our experiments, the normal map looks correct and does not guarantee the correct reconstruction of the surface by marching cubes, especially when the normal map has noise in the details.
>
> **R4-Q4. Discretization formulas and multiple surface intersections.**
>
> For the discretization, one can bring Eq.5 and Eq.6 into Eq.7, and take advantage of the properties of the derivative of the logistic sigmoid function $\Psi'_s =s\Psi_s(1-\Psi_s)$. We can get the $\sigma$ formula for the discretization computation:
> \begin{equation}
> \sigma ({\bf{r}}(t_i)) = s\left(\Psi\left( {f\left( {{\bf{r}}(t_i)} \right)} \right) -1 \right)\nabla f\left( {{\bf{r}}(t_i)} \right) \cdot \bf{d}
> \end{equation}
>
> Then the volume rendering integral can be approximated using $\alpha$-composition, where $\alpha_i = 1 - exp \left(-{\sigma_i} \left({t_{i+1}} - {t_i}\right)\right)$. For multiple surface intersections, we follow the same strategy as NeuS, where $\alpha_i = clamp\left( {\alpha_i,0,1} \right)$.
>
> **R4-Q5. It is not certain how the displacement constraint is imposed, nor its relation to equation 13**
>
> We found that equation 13 of the original paper has a typo. We revised it with the following form. The displacement constraint is simply used to multiply the implicit displacement.
> \begin{equation}
> f({\bf{x}}) = {MLP_b}({\gamma }({\bf{x,\alpha}}_b) - 4\Psi{'_s}(f_b) {MLP_d}\left( {{\gamma}({\bf{x,\alpha}}_d)} \right){\bf{n}}),
> \end{equation}

---

> > ### Author Response · Authors · 2022-08-02
> > **Comments**
> >
> >
> > **R4-Q6.  Could the authors explain their choice of the logistic sigmoid function for Psi? such as the Gaussian or Laplace distributions CDF.**
> >
> > The CDF of the Gaussian distribution is not an explicit elementary function. This approach requires integration in a discrete approximation that introduces additional error.
> > Using the sigmoid function allows for more efficient derivation of density formulas and reduces numerical problems caused by division. We add a comparison in the revision. In Fig. 8, "OUR Base-Laplace" means using CDF of Laplace distributions for $\Psi$. In the negative semi-axis, the derivative of the Laplace distribution CDF divided by the distribution is a constant function, the scale parameter $s$, and this will affect the quality of surface reconstruction.
> >
> > **R4-Q7. I suspect equation 9 has a typo, and it should be $f_d(x_b)$?**
> >
> > We previously assumed that the displacement function between the basis function point and the combined surface is the same. But the problem is two displacement functions are not the same. In order to explain this more clearly, we define the displacement implicit function $f_{d'}$ to map the point $x_b$ on the base surface to the surface point $x$ along the normal $n_b$ and $f_{d}$ is used to map the point $x$ on the base surface to the surface point $x_b$ along the normal $n_b$, thus $f_{d'}(x_b) = f_d(x)$. We fixed this in the revision.
> >
> > **R4-Q8. The intention behind equation 15 is unclear. Could the authors please explain?**
> >
> > For the signed distance field, the norm of the gradient is expected to be 1.
> > If the norm is greater than 1, it means that the signed distance at that location changes more drastically, which means that the change of signed distance function is larger in a shorter distance.
> > We propose to use the norm of the gradient of the signed distance field to weight the parameter $s$ in these short distances. We increase $s$ when the norm of the gradient along the ray direction is larger than 1 to further consider these regions to improve. Because we only need to guarantee the accurate SDF at a 0 level set to extract the surface, we need to find the gradient error near the surface, so we use normalized $\Psi'_s$ as a weighting function.
> >
> > **R4-Q9. I encourage the authors to move the limitations section to the main paper in the revised version.**
> >
> > Thanks for your suggestion, we will move the limitation section to the main paper.

---

> ### Author Response · Authors · 2022-08-08
> **The completeness of the author's response**
>
> Dear reviewer, we are very thankful for your suggestions and help in improving several parts of the paper. And if that's possible, it would be crucial for us to know if there are any concerns left on your side after our response.  We would be glad to know whether there is anything else we could elaborate on to address existing or any new concerns.

---

### Official Review · Reviewer_AK1b · 2022-07-08

**Rating:** 6
**Confidence:** 4
**Soundness:** 3 good
**Presentation:** 2 fair
**Contribution:** 2 fair

**Summary:**


 This paper builds on the recent efforts in multi-view neural implicit surface reconstruction and improves in the reconstruction quality by introducing a different way of modelling SDFs, applying the implicit displacement fields to this task, and also proposing locally-adaptive scaling factors. They show decent improvements on multiple object-level dataset for both surface reconstruction and view synthesis.


**Questions:**

- I wonder if you simply change the way of modelling SDFs of NeuS to yours, how much benefit can you obtain (without the displacement field) for NeuS?


 - No enough acknowledgement to the IDF paper in 3.2. For example, $\mathbf{x}_b = \mathbf{x} - f_d(\mathbf{x}_b)\mathbf{n}_b$ (the key idea of implicit displacement field) mentioned already in IDF.

 - Why not do the same as in the IDF paper where they use two SIRENs for the base model and displacement model but using different frequencies?


 - PSNR calculation. Do you calculate the PSNR on the training views or the held-out testing views?

 - Ablation study. Table 3 confuses me since it lacks of discussion of different componenets. Why Base+H is much worse than only Base? Why changing from only base to IDF does not gurantee better results (DTU experiment)? And it seems to me that most beneficial components are coming from the coarse-to-fine strategy and the locally-adaptive scales.



**Limitations:**

Limitations and societal impact are briefly discussed in the conclusion.

**Strengths And Weaknesses:**

*Strengths*
 - I like the idea of locally-adaptive scaling factor s, which makes a lot of sense and I think can be a general add-on for other methods.
 - The paper is well written overall, easy to follow.

 *Weaknesses*
- In 3.1, what is the motivation of modelling SDFs with transparency? I don't see the clear advantage of doing so compared to modelling with the density function as done in VolSDF, or weighting function in NeuS. Indeed, modelling with the transparency is indeed conceptually simper than NeuS, but fundamentally it is very similar to the combination of NeuS and VolSDF.
- I don't think the method has fully utilize the benefit of implicit displacement field. For example, intuitively speaking, using a base model to model the coarse shape and another network to model the details should speed up the optimization process a lot, since the coarse shape can be very quickly obtained, and learning displacements is an easier optimization task than learning fully SDFs. Any experiments show the runtime benefit?
- According to both quantitative and qualitative results, even after adding 3 different contributions and course-to-fine strategies, the proposed approach is only marginally better. I am not fully convinced by the usefulness of your different contributions. Moreover, because of the use of an extra MLPs, you even need more computational resources.

---

> ### Author Response · Authors · 2022-08-02
> **comment**
>
> We thank the reviewer for the insightful and detailed review. We respond to each question in the following.
>
> **R3-Q1. What are the advantages of modeling SDFs with transparency?**
>
> Please refer to ALL-Q1.
>
> **R3-Q2. Should IDF speed up the optimization process? Any experiments show the runtime benefit of IDF?**
>
> Although training an IDF is easier than training an SDF, the introduction of an additional MLP results in a longer training time for a single iteration. So overall, it does not improve the runtime efficiency of the optimization process, but it can get more detailed reconstruction results. The training time of each scene is around 20 hours for 300k iterations. We will add this to the experiment section in the revision.
>
> **R3-Q3. According to both quantitative and qualitative results, the proposed approach is only marginally better.**
>
> Since the error of the high-frequency details is small, the overall Chamfer distance is not greatly improved. However, this is relative. The methods for 3D reconstruction are well refined and generally have very good results. Compared to the improvements that are typical in the field, our improvements are significant.
>
> In addition, on the visual side, we provide a heatmap result of the local errors in Fig.12 of the revised supplemental material to show the significant improvement visually. It can be seen that we have a higher improvement in the details, such as the roof and the details in the shovel of the excavator. Furthermore, compared with the IDR[31], VolSDF and NeuS have only an improvement of less than 0.1 on the DTU dataset in their paper. Our improvement is also 0.1.
>
> **R3-Q4. How much benefit can you obtain (without the displacement field) for NeuS?**
>
> Please refer to common comment ALL-Q1. Note that we also provide a comparison with NeuS and VolSDF in Fig. 8 of the supplementary material. Our "OUR Base-Sigmoid" result is our method without the displacement field, which is better than the competitors.
>
> **R3-Q5. Not enough acknowledgment of the IDF paper in Section 3.2. For example, $x_b=x-f_d(x_b)n_b$ the key idea of implicit displacement field mentioned already in IDF.**
>
> Please refer to common comment ALL-Q2. Note that we have cited [32] and discussed it in Section 3.2 now.
>
> **R3-Q6. Why not do the same as in the IDF paper where they use two SIRENs for the base model and displacement model but using different frequencies?**
>
> Please also refer to common comment ALL-Q2. Note that we provide an experiment to compare our method and the method used in [32] in Fig. 8 of the revised supplementary material to show the benefits of increasing frequency gradually, where "OURS-Siren" is the method used in [32].
>
> **R3-Q7. Do you calculate the PSNR on the training views or the held-out testing views?**
>
> All PSNR results are on the training views.
>
> **R3-Q8. Why Base+H is much worse than only Base? Why changing from only base to IDF does not guarantee better results (DTU experiment)? And it seems to me that the most beneficial components are coming from the coarse-to-fine strategy and the locally-adaptive scales.**
>
> The improvement of the geometric fidelity using high-frequency details is challenging, not only because the high-frequency error is small, but also because the high-frequency noise will affect the further improvement of the fidelity. We provide a visualization of the ablation as other reviewer requested in Fig. 11 of the revised supplementary material. We show that directly using Base+H will destroy the reconstruction of geometry, even though the image quality is great. The IDF+H can alleviate some of the high-frequency noise, but the result has a large geometric error in some regions. We propose the method of gradually adding frequency to the learning processing, which can reconstruct a fine surface while suppressing noise.
>
> **R3-Q9. Limitations and societal impact are briefly discussed in the conclusion.**
>
> Please refer to the common comment ALL-Q3.

---

> ### Comment · Reviewer_AK1b · 2022-08-08
> **Response to authors' rebuttal**
>
> Thanks for providing detailed responses to my questions!
>
> I appreciate the clarification of the differences between your approach and the IDF paper, the motivation of using positional encoding instead of two SIREN networks, and also give acknowledgement to IDF in the method part. Also, I appreciate your explanation on the ablation study and would be great if you incorporate them in the final paper (main paper or supp. mat.).
>
> From the rebuttal to other reviewers' and my comments, I would upgrade my scores accordingly to accept.

---

> > ### Author Response · Authors · 2022-08-08
> > **Response**
> >
> > Thank you for your feedback and suggestions. We will incorporate all the explanations and discussions in the final version of the main paper or supplemental material.

---

### Official Review · Reviewer_t73h · 2022-07-10

**Rating:** 6
**Confidence:** 4
**Soundness:** 3 good
**Presentation:** 3 good
**Contribution:** 2 fair

**Summary:**

The authors address the objective of improving the accuracy of 3D scene reconstruction based on neural rendering approaches. For this purpose, they first provide a theoretical discussion on how the signed distance function (SDF) can be embedded into volume rendering as well as the relationship between the SDF, the transparency function, the volume density and the weighting function. In addition, criteria for functions that are suitable to map signed distances to transparency and a respectively suitable class of functions is discussed, but the choices follow previous work (in particular, the integration of the SDF follows VolSDF, and the selection of a logistic sigmoid function with learnable scale parameter for Psi follows NeuS). This is used in a multi-scale fitting framework where the SDF is modeled as a combination of a base distance function and a (implicit) displacement function (similar to references [reference 15] and [reference 32]) along the normal of the base distance function, and positional encoding is used for each of these functions separately, where the frequency of the positional encoding is increased according to the coarse-to-fine strategy by Park et al. [reference 22]. Both functions are additionally represented by separate MLPs, and the used loss includes a radiance term and an additional Eikonol regularization of the SDF, where the norm of the gradient of the SDF is used to weight the scale parameter in a spatially varying manner.

Experiments on the DTU Dataset, NeRF Synthetic Dataset and BlendedMVS dataset with comparisons to NeuS and VolSDF show the potential of the proposed approach.

**Questions:**

## Exposition:
- As mentioned above under 'strengths/weaknesses', some selections of reference lists seem arbitrary. For NeRF improvements, it might be helpful to cluster references regarding the aspect the techniques are focused on, e.g. faster training/inference, less constrained conditions, video inputs, generalization capabilities, multi-scale representations, etc.. The lists of papers for 3D reconstruction and surface reconstruction also seem arbitrary in the current formulation. Referring to the respective works or respective surveys could solve this.

## Evaluation:
- I would recommend to also provide the equation for the Chamfer distance, as there are different definitions used in literature.
- Regarding Figures 1 and 3, showing the local errors on the surface with respect to the ground truth might additionally better highlight where deviations are larger.
- As mentioned above, the MipNeRF approach focuses on scene representation at continuously-valued scale which improves the representation of fine-grained details. Why has this approach has only been mentioned in the list in the introduction, but not in the discussion of related work or among the competing techniques used for evaluation.
- As mentioned above, training and inference times could be mentioned.
- As mentioned above, an ablation study of the effect of the regularizers would be interesting as well.
- As mentioned above, the stability of the reconstructions of fine-grained structural details could be shown with respect to the number of views. Maybe the proposed approach gets the same quality of the reconstruction of NeRF with significant less input views.
- As mentioned above, there is no detailed discussion of problematic cases or failure cases in the main paper. Limitations regarding inaccurate reconstructions of linear structures (i.e. ropes) mentioned in the supplemental should rather be mentioned in the main paper. In addition, e.g., for the cases where the proposed approach does not outperform its competitors in Table 1, respective visualizations could be provided to demonstrate the kind of errors and where these are larger than for the other methods. This would provide insights on further improvement potential.
- A discussion or comparison to the recent approaches Instant-NGP (Muller et al., Instant neural graphics primitives with a multiresolution hash encoding) and SVLF (Tremblay et al., RTMV: A Ray-Traced Multi-View Synthetic Dataset for Novel View Synthesis) would be interesting, however, it seems as if these works were not yet published at the NeuRIPS submission deadline.

## Reproducibility:
- Will code be release upon paper acceptance?

**Limitations:**

The authors have adequately addressed the potential negative societal impact of their work. Limitations in terms of the need for optimizing an additional implicit function or overfitting have only been shortly discussed in the last section on conclusions. As mentioned above, the discussion of limitations needs to be improved.

**Strengths And Weaknesses:**

## Originality/novelty and technical soundness:
pro:
+ The approach seems reasonable.
+ The major novelty seems combining these concepts in the NeRF framework and using a spatially varying scale parameter which, together with the combination of all the concepts, leads to accuracy improvements over the other approaches (NeuS, VolSDF) in the comparison.

con:
- However, there is a larger overlap with existing work regarding the use of separate networks to represent a base SDF and displacement fields [references 15 and 32], the integration of the SDF into volume rendering [reference 30], using an unbiased density function using logistic sigmoids and learnable control parameter [reference 28]).



## Clarity and exposition:
pro:
+ The paper is well-structured and easy to follow. Arguments are clear.
+ Figures/Tables and the respective captions are informative.
+ The relevance of the topic is sufficiently motivated.
+ The supplemental provides further implementation details in more detail.

con:
- While text quality is mostly ok, there are typos, e.g. 'view-dependent' appears several times as 'view dependent', 'low/high frequency' should be written as 'low/high-frequency' if it is followed by a noun it refers to (e.g., 'components', 'content', 'surfaces', etc.), 'state of the art algorithm' should read 'state-of-the-art algorithm', 'coarse to fine optimization' should read 'coarse-to-fine optimization'.

- Some selections of reference lists seem arbitrary, e.g., the reference lists on NeRF improvements (in the introduction) and multi-view stereo reconstruction (in the related work section). It is not clear why only these references have been used. For NeRF improvements, it might be helpful to cluster references regarding the aspect the techniques are focused on, e.g. faster training/inference, less constrained conditions, video inputs, generalization capabilities, multi-scale representations, etc.. Regarding voxel-based 3D reconstruction approaches there are many more variants, e.g.:
- - Izadi et al., KinectFusion: real-time 3D reconstruction and interaction using a moving depth camera, and its extensions such as:
- - Niessner et al., Real-time 3D reconstruction at scale using voxel hashing
- - Whelan et al., Kintinuous: Spatially extended kinectfusion
- - Dai et al., Bundlefusion: Real-time globally consistent 3d reconstruction using on-the-fly surface reintegration
- - Klingensmith et al., Chisel: Real Time Large Scale 3D Reconstruction Onboard a Mobile Device using Spatially Hashed Signed Distance Fields
- - Stotko et al., Efficient 3D Reconstruction and Streaming for Group-Scale Multi-Client Live Telepresence

In addition, there are more variants for surface meshing such as:
- - Fuhrmann and Goesele, Floating Scale Surface Reconstruction
- - Calakli and Taubin, SSD: Smooth signed distance surface reconstruction
- - Berger et al., A survey of surface reconstruction from point clouds



## Evaluation:
pro:
+ The authors provide qualitative and quantitative results with comparisons to reasonable baselines (VolSDF and NeuS), where the proposed approach is demonstrated to lead to improvements in the reconstruction result.
+ An ablation study shows the effects of high-frequency positional encoding, coarse-to-fine optimization and the use of the implicit displacement function.
+ The supplemental provides further ablations on the coarse-to-fine parameter alpha_d and the number of frequency bands L as well as ablations for adaptive transparency.

con:
- The MipNeRF approach that focuses on scene representation at continuously-valued scale also improves the representation of fine-grained details. It is not clear why this approach has only been mentioned in the list in the introduction, but not in the discussion of related work or among the competing techniques used for evaluation.
- Training and inference times could be mentioned.
- An ablation study of the effect of the regularizers would be interesting as well.
- Additionally, the stability of the reconstructions of fine-grained structural details could be shown with respect to the number of views. Maybe the proposed approach gets the same quality of the reconstruction of NeRF with significant less input views.
- There is no detailed discussion of problematic cases or failure cases in the main paper. Limitations regarding inaccurate reconstructions of linear structures (i.e. ropes) mentioned in the supplemental should rather be mentioned in the main paper. In addition, e.g., for the cases where the proposed approach does not outperform its competitors in Table 1, respective visualizations could be provided to demonstrate the kind of errors and where these are larger than for the other methods. This would provide insights on further improvement potential.



## Reproducibility:
pro:
+ The information in the paper together with the attached code seems sufficient to re-implement the approach.

con:
- It is not entirely clarified, whether the code will also be made available afterwards.



## References:
con:
- See comment under 'Exposition'.
- For several of the references, only the arxiv reference is provided and not the final conference/journal reference. This should be revised accordingly.



## Post-rebuttal recommendation:
The paper addressed my major concerns in terms of ...
... the detailed discussion of the works of VolSDF [30], NeuS [28] and Yifan et al.'s approach [32] which helps to better see the embedding of the proposed approach in the context of related work,
... the improvements regarding the exposition (typos, also mentioning MipNeRF in the related work, mentioning the used Chamfer distance formula, etc.),
... the promised code release,
... the additional details regarding training/inference times,
... the shift of limitations into the main paper,
... the added visualization of local errors.
As a result, I increase my rating from borderline accept to weak accept.

---

> ### Author Response · Authors · 2022-08-02
> **comments**
>
> We thank the reviewer for the insightful and detailed review. We respond to each question in the following.
>
> **R2-Q1. What is the Novelty for IDF compared with [15, 32] and SDF calculation compared with [28] and [30]?**
>
> Please refer to the answers in ALL-Q1 and ALL-Q2.
>
> **R2-Q2. Typos.** Thanks for pointing out the typos. We have fixed them in the revision.
>
> **R2-Q3. Some selections of reference are arbitrary and some reference is missing or arxiv version.**
>
> Thanks for the suggestion. Our related work focuses on the recent neural implicit surface reconstruction. We will add the related voxel-based 3D reconstruction references as suggestions and improve the writing to extend the NeRF part to the revision. We also have used the published format instead of arxiv version in the revision if available.
>
> **R2-Q4. Why has MipNeRF only been mentioned in the list in the introduction, but not in the discussion of related work or among the competing techniques used for evaluation?**
>
> Thanks for the suggestion, we will discuss MipNeRF in the detail reconstruction section and expand the recent work of NeRF in related work. Both MipNeRF and NeRF focus on density reconstruction and cannot guarantee to produce watertight surfaces like SDF.  We could try to evaluate the MipNeRF, but it is generally assumed that pure image-based NeRFs cannot compete with NeRFs that have a specific surface regularizer when it comes to surface reconstruction. We would assume that MipNeRF is better in terms of PSNR, but worse in terms of surface construction. The main issue is that extracting surfaces from MipNeRF needs extra code in Jax and we currently use PyTorch.
>
> **R2-Q5. Time of training and inference.**
>
> The training time of each scene is around 20 hours for 300k iterations. The inference time for extracting a mesh surface with high resolution (512 grid resolution for marching cubes) is around 60 seconds and rendering an image at the resolution of 1600x1200 is around 540 seconds. We will add this to the experiment section in the revision. This setting of resolution is similar to other approaches for surface reconstruction (i.e. NeuS).
>
> **R2-Q6. An ablation study of the effect of the regularizers would be interesting as well.**
>
> We provide an ablation study of Eikonal regularization in Fig.8 in the revision of the supplemental material. We observe that training without the regularization of the base SDF ("OUR-w/o Base Reg") results in slightly worse reconstruction quality. Thus constraining the base SDF can help improve the quality of the reconstruction.
>
> **R2-Q7. The stability of the reconstructions of fine-grained structural details could be shown with respect to the number of views.**
>
> Thanks for the suggestion. We conduct an experiment for surface reconstruction with 10\% of the training image in Fig.9. We find that our method can keep the structure of reconstructed objects complete compared to NeuS, and can better reconstruct parts such as thin stripes with fewer training images. PSNR of NeuS and OURS is 28.31 and 31.77 respectively, which is also improved.
>
> **R2-Q8. Limitations should be discussed in the main paper and the visualizations should be provided for bad cases.**
>
> Thanks for the suggestion. Please also refer to ALL-Q3. We extend the limitation and visualize a bad case of Table 1 where the error is larger than that of the other methods. In this case, the lighting of this model varies and the texture is not as pronounced, thus it is difficult to reconstruct the details of the belly.
>
> **R2-Q9. Will code be released upon paper acceptance?**
>
> We will release the code upon paper acceptance.
>
> **R2-Q10. The equation of the Chamfer distance.**
>
> We added the formula of Chamfer distance in the supplemental material B.2 in the revision.
>
> **R2-Q11. Showing the local errors on the surface with respect to the ground truth to highlight where deviations are larger.**
>
> Thanks for your suggestion. We added a visualization for local errors in Fig.12 of the revised supplemental material to better highlight the local errors. It can be seen that we have a higher improvement in the details, such as the roof and the details in the shovel of the excavator.

---

> > ### Comment · Reviewer_t73h · 2022-08-08
> > **Response to Author Feedback**
> >
> > Thanks for providing detailed comments on the different concerns.
> >
> > In particular, I appreciate the adequate discussion of ...
> > - ... the detailed discussion of the works of VolSDF [30], NeuS [28] and Yifan et al.'s approach [32] which helps to better see the embedding of the proposed approach in the context of related work,
> > - ... the improvements regarding the exposition (typos, also mentioning MipNeRF in the related work, mentioning the used Chamfer distance formula, etc.),
> > - ... the promised code release,
> > - ... the additional details regarding training/inference times,
> > - ... the shift of limitations into the main paper,
> > - ... the added visualization of local errors.

---

> > > ### Author Response · Authors · 2022-08-08
> > > **Response**
> > >
> > > Thank you for your response. It would be crucial for us to know if there are other concerns left on your side. We will incorporate all the changes in the final paper.

---

### Official Review · Reviewer_XTFd · 2022-07-11

**Rating:** 6
**Confidence:** 4
**Soundness:** 3 good
**Presentation:** 4 excellent
**Contribution:** 2 fair

**Summary:**

The paper tackles the task of surface reconstruction from images. They use the volume rendering based approach popularized by NeRF, and thus are also interested in maintaining high novel view synthesis performance. In particular they follow recent work (IDF, VolSDF) that embed an SDF into the volume rendering pipeline in order to get more accurate surfaces, and propose an approach for achieving high frequency detail. To do this, they incorporate the SDF in a new fashion, use implicit displacement fields [32] and use a coarse to fine strategy.

**Questions:**

In section 3.2 the work describing the decomposition into a base and displacement field is not cited at all (should be [32]). Given that it is a major part of the approach and the use of that work is not cited anywhere except in related work, this is quite important to do.

**Limitations:**

Both limitations and potential negative societal impacts have been discussed in the conclusion.

**Strengths And Weaknesses:**

Strengths
- Good explanation of previous ways that SDFs have been incorporated into volume rendering
- Good explanation for the proposed way to incorporate SDFs and the intuition behind it.
- Experimented on well established datasets/benchmarks and shows significant improvements on surface reconstruction while maintaining similar or better view synthesis quality
- Good ablation for the high frequency detail methods, e.g. basic encoding vs displacement field method, and the benefit of the coarse to fine approach.

Weaknesses
- No theoretical explanation of the benefits of the new SDF incorporation method compared to previous methods, except for simplicity of the derivation. You have done an excellent job motivating your approach mathematically, but haven't clearly explained what benefits that has over the other methods. You should explain in more detail why directly modelling the transparency function would be better.
- No ablations showing how the new way to incorporate SDFs differs in performance to the way VolSDF/NeuS does it.

---

> ### Author Response · Authors · 2022-08-02
> **Commont**
>
> We thank the reviewer for the insightful and detailed review. We respond to each question in the following.
>
> **R1-Q1. Why directly modeling the transparency function would be better?**
>
> Compared to VolSDF, since the transparency function is explicit, our method can use an inverse distribution sampling computed with the inverse CDF to satisfy the approximation quality. Thus no complex sampling scheme as in VolSDF is required. Compared with NeuS, we obtain a simpler formula for the density $\sigma$ for discretization computation, reducing the numerical problems caused by division using in the NeuS. Our approach does not need to involve two different sampling points, namely section points and mid-points, where the color and the geometry are more consistent. Please refer to ALL-Q1 that we discuss this with theoretical explanation in the common comment ALL-Q1.
>
>
> **R1-Q2. No ablations showing the difference of performance compared with VolSDF/NeuS.**
>
> We provide the experiment as requested in Fig. 8 "OUR Base-Sigmoid". Due to the easy approximation, no numerical problems due to division and no need to sample section points and mid-points separately, the result shows better geometry consistency and quality.
>
> **R1-Q3. Why does the use of the work[32] is not cited anywhere except in related work.**
>
> Thanks for the suggestion of citation. We also discussed this in the common comments. Please see the answer in ALL-Q2.

---

> > ### Comment · Reviewer_XTFd · 2022-08-09
> > **Reply to the Author's Response**
> >
> > Thank you for the response. My questions have been sufficiently addressed and some of my feedback has been integrated into the revised paper.
> >
> > I have also read the other reviews and the authors' response to those papers, and have no further questions.
> >
> > I stand with my original rating and recommend accepting the paper.

---

### Author Response · Authors · 2022-08-02
**Common comments**

We thank all the reviewers for the thorough and constructive reviews. In the following we first address the common concerns.

**ALL-Q1. Can you provide more details about the advantages of the proposed modeling of the transparency function compared with VolSDf and NeuS?**

Compared to VolSDF, since the transparency function is explicit, our method can use an inverse distribution sampling computed with the inverse CDF to satisfy the approximation quality. Thus no complex sampling scheme as in VolSDF is required.


Compared with NeuS, we obtain a simpler formula for the density $\sigma$ for the discretization computation, reducing the numerical problems caused by division in NeuS. Bringing Eq.5 and Eq.6 into Eq.7, we get the $\sigma$ formula for the discretization.
\begin{equation}
\sigma ({\bf{r}}(t_i)) = s\left(\Psi\left( {f\left( {{\bf{r}}(t_i)} \right)} \right) -1 \right)\nabla f\left( {{\bf{r}}(t_i)} \right) \cdot \bf{d}
\end{equation}
Then the volume rendering integral can be approximated using $\alpha$-composition, where $\alpha_i = 1 - exp \left(-{\sigma_i} \left({t_{i+1}} - {t_i}\right)\right)$.
Furthermore, our approach does not need to use two different sampling points, namely section points and mid-points used in NeuS, which makes it easier to satisfy the unbiased weighting function. Since there is no need to calculate the SDF and the color separately for the two different point sets, the color and the geometry are more consistent compared to NeuS. We will add this discussion to the main text in the revision. In Fig. 8 of the supplementary material, we also provide a qualitative comparison to Volsdf and NeuS requested by the reviewers. For example, our "Base-Sigmoid" result shows better geometry consistency on the roof of the house compared with VolSDF and NeuS.

**ALL-Q2. What exactly is novel compared with [32]? Why is [32] only discussed in the related work?**

Thanks for the suggestion of citing [32] in Section 3.2. We have cited [32] and discussed in Section 3.2 now.

We would like to note some differences to [32]. We use positional encoding instead of Siren, so that the frequency can be explicitly controlled by a coarse-to-fine strategy better than simply using two Siren networks with two different frequency levels. This is very useful when 3D supervision is not given. We provide results in Fig.8 of the supplementary material to answer the difference raised by the reviewers.
We observed that the IDF using Sirens ("OURS-Siren" in Fig.8) used in [32] can obtain a high PSNR result but low geometry fidelity. Although [32] also use a coarse-to-fine strategy between two frequency levels, we found that the method still has the problems when learning high-frequency details because of the high-frequency noise involved at the beginning. Our IDF using positional encoding does not use high-frequency information at the beginning of training, which makes the training more stable. In general, we provide a solution that allows more fine-grained control over frequency. This approach is more stable for the case without 3D supervision.


**ALL-Q3. There is no detailed discussion of limitation in the main paper rather than in the supplementary material.**

Thanks for the suggestion. We will move some of the limitation part to the main text of the paper.

---

### Meta-Review · Area_Chair_VkLV · 2022-08-25

**Recommendation:** Accept
**Confidence:** Certain

**Metareview:**

This paper presents a new way to build centered weights for volume rendering, utilize displacement maps, adaptive scale, as well as other techniques to provide better high frequency details in neural SDF representations.
The reviewers also acknowledged the rebuttal and the revision and the authors addressing their main concerns.

**Award:**

No

---

### Decision · Program_Chairs · 2022-09-14

Accept